# Mechanistic classification and benchmarking of polyolefin depolymerization over silica-alumina-based catalysts

Wei-Tse Lee[1], Antoine van Muyden [1], Felix D. Bobbink[1], Mounir D. Mensi[1], Jed R. Carullo[1] & Paul J. Dyson [1]✉

Carbon-carbon bond cleavage mechanisms play a key role in the selective deconstruction of alkanes and polyolefins. Here, we show that the product distribution, which encompasses carbon range and formation of unsaturated and isomerization products, serves as a distinctive feature that allows the reaction pathways of different catalysts to be classified. Co, Ni, or Ru nanoparticles immobilized on amorphous silica-alumina, Zeo-Y and ZSM-5, were evaluated as catalysts in the deconstruction of *n*-hexadecane model substrate with hydrogen to delineate between different mechanisms, i.e., monofunctional- (acid site dominated) or bifunctional-hydrocracking (acid site & metal site) versus hydrogenolysis (metal site dominated), established from the product distributions. The ZSM-5-based catalysts were further studied in the depolymerization of polyethylene. Based on these studies, the catalysts are plotted on an activity-mechanism map that functions as an expandable basis to benchmark catalytic activity and to identify optimal catalysts that afford specific product distributions. The systematic approach reported here should facilitate the acceleration of catalyst discovery for polyolefin depolymerization.

Valorization of end-of-life plastic materials continues to receive significant attention due to emerging environmental concerns and public awareness[1-3]. Major post-treatments of waste plastics include recycling (~11%), incinerating (~14%), and landfilling (~75%) among the 380 M tons total annually produced[4]. More sustainable approaches are foreseen with a significantly increased proportion of recycling[5]. Polyolefins (polyethylene, PE, polypropylene, PP, polystyrene, PS, etc.) comprise more than 50% of the global synthetic polymers produced[6], and their transformation to short-medium carbon range chemicals, e.g., gases ($C_{1-4}$), gasoline ($C_{5-12}$), diesel ($C_{12-20}$), or waxes ($C_{18-36}$), are desired pathways either for direct applications as energy carriers[7-9] or as circular feedstocks for the chemical industry[10-13]. Studies have focused on the chemical conversion of saturated polyolefins, especially PE, given its considerable quantity and typical application in single-use consumer products[14,15].

Various approaches to depolymerize polyolefins have been reported. Thermal cracking using solely heat under an oxygen-absent atmosphere takes place at temperatures of 550–600 °C typically, though lower temperatures (~400 °C) can be applied in the presence of a catalyst[16,17]. For instance, Figueiredo et al. reported the conversion of low-density PE (LDPE) to afford gaseous ($C_{1-4}$) and liquid ($C_{5-20}$) hydrocarbons in 46 and 44% yield, respectively, including isomerization and unsaturated products, via monofunctional cracking over ZSM-5 at 400 °C (Table 1, Entry 1)[18]. Munir et al. achieved near-quantitative conversion of mixed plastics (PE, PP, and Polystyrene, PS) with gaseous and liquid hydrocarbons obtained in 36 and 59% yield, respectively, via monofunctional hydrocracking using Beta zeolite under $H_2$ (20 bar) at 400 °C (Table 1, Entry 2)[19]. Jumah et al. reported the depolymerization of LDPE to gaseous and liquid hydrocarbons in 44 and 52% yield, respectively, including isomerization products, via

[1]Institute of Chemical Sciences and Engineering, École Polytechnique Fédérale de Lausanne (EPFL), Lausanne, Switzerland. ✉e-mail: paul.dyson@epfl.ch

**Table 1 | Plastic material depolymerizations via monofunctional-, bifunctional- hydrocracking, tandem catalysis, and hydrogenolysis pathways**

| Entry | Sub. | Catalyst | S/C ratio* | Type | Temp. (ºC) | Time (h) | Press. (bar) | Conv. (%) | Yield† (Gas, %) | Yield† (Liq., %) |
|---|---|---|---|---|---|---|---|---|---|---|
| 1 | LDPE[a] | ZSM-5 | 100 | Monofunctional cracking | 400 | 2 | N/A | 90 | 46 | 44 |
| 2 | Mixed[b] | Beta | 20 | Monofunctional hydrocracking | 400 | 1 | 20 | 95 | 36 | 59 |
| 3 | LDPE[c] | Pt_1 wt%/Beta | 10 | Bifunctional hydrocracking | 330 | 1 | 20 | 99 | 44 | 52 |
| 4 | LDPE[d] | Pt/WO$_3$/ZrO$_2$ + Zeo-Y | 10 | Tandem catalysis | 250 | 2 | 30 | 99 | 9 | 83 |
| 5 | LDPE[e] | Ru/C | 28 | Hydrogenolysis | 225 | 16 | 20 | 99 | 55 | 45 |
| 6 | HDPE[f] | Ru/C | 2 | Hydrogenolysis | 220 | 1 | 60 | 99 | 12 | 68 |
| 7 | PP[g] | Ru/TiO$_2$ | 20 | Hydrogenolysis | 250 | 16 | 30 | >94 | 28 | 66 |

*S/C ratio substrate/catalyst weight ratio; Note that the selected carbon range for the reported liquid hydrocarbon yields (wt%) may have slight differences among literature.
†Gaseous products are typically defined as C$_{1-4}$.
‡Liquid products are typically defined as C$_{5-20}$.
[a]Plastic properties/source not provided.
[b]HDPE (ρ = 0.952 g/cm$^3$, Sigma-Aldrich) + LDPE (ρ = 0.918 g/cm$^3$, Sigma-Aldrich) + PP (Mw = 250,000 g/mol, ρ = 0.9 g/cm$^3$) + PS (Mw = 192,000 g/mol) = (40 + 10 + 30 + 20)wt%.
[c]LDPE (Mw ~150,000 g/mol).
[d]LDPE (Mw ~250,000 g/mol, Sigma-Aldrich).
[e]LDPE (Mw ~4000 g/mol, Sigma-Aldrich).
[f]HDPE water jug from local source, properties not specified.
[g]Isotactic PP (Mw ~250,000, Sigma-Aldrich).

bifunctional hydrocracking employing Pt_1 wt%/Beta as catalyst under H$_2$ (20 bar) at 330 °C (Table 1, Entry 3)[20]. In addition to hydrocracking via silica-alumina materials, tandem catalysis approaches combining precious metal clusters immobilized on other solid acid supports (i.e., not silica-alumina structure) have also been reported. For example, Liu et al. were able to convert LDPE to gaseous and liquid hydrocarbons in 9 and 83% yield, respectively, by applying 1 wt% Pt loaded on WO$_3$/ZrO$_2$ and Zeolite-Y under H$_2$ (30 bar) at 250 °C (Table 1, Entry 4)[21]. However, the tandem catalysis has not yet been resolved in detail or struck by a consensus as to the silica-alumina-based catalysts and, thus, will not be intensively studied and incorporated in our following studies.

Hydrogenolysis of C-C bonds using Ru-based catalysts has also been used for the selective deconstruction of plastics, limiting isomerization and unsaturated products. The formation of methane is considered an important indicator of a hydrogenolysis mechanism with alkylidyne intermediates, as C$_1$ intermediates such as the methenium cation are disfavored in (hydro)cracking mechanisms[22]. Ru NPs on activated charcoal (Ru/C) has been shown to cleave PE into shortened hydrocarbons under H$_2$ (20–60 bar) at 200–225 °C[23,24]. Julie et al. reported quantitative conversion of LDPE to gaseous and liquid hydrocarbons in 55 and 45% yield, respectively, using Ru_5wt%/C under H$_2$ (20 bar) at 200 °C (Table 1, Entry 5)[23]. The same catalyst was shown to transform high-density PE (HDPE) into gaseous and liquid hydrocarbons in 12 and 68% yield, respectively, under H$_2$ (60 bar) at 220 °C in hexane (Table 1, Entry 6)[24]. Ru NPs immobilized on a reducible solid support such as TiO$_2$ have also been reported to catalyze PP hydrogenolysis (Table 1, Entry 7)[25]. Pt NPs immobilized on metal oxide (SrTiO$_3$) or fabricated into a mesoporous shell/active site/core structure (mSiO$_2$/Pt/SiO$_2$) have also been used in the hydrogenolysis of PE to afford wax-lubricant range products (C$_{18-40+}$) under 9–14 bar H$_2$ at temperatures ranging from 250 to 300 °C reacting for 24–96 h[26–28].

Despite the abundant literature describing catalysts that deconstruct alkanes and polyolefins, systematic comparisons among the catalysts are largely absent and their mechanistic pathways are often overlooked, which hinders further rational catalyst design. To overcome these limitations, we studied the catalytic deconstructions of n-hexadecane (nC$_{16}$−used as a model substrate and functioning as a benchmark reaction) and polyethylene (PE, Mw ~4000 g/mol) in the presence of hydrogen using various heterogeneous catalysts prepared using an identical synthetic procedure. We use the product distribution, including the carbon range and distinctive features such as

degrees of saturation and the formation of isomerization products, were evaluated[22], to classify the C-C bond cleavage mechanisms as:

1. Monofunctional hydrocracking−acid site dominated cleavage with a unimodal distribution comprising C$_{3-4}$ as major products including significant isomerization and unsaturated products.
2. Bifunctional hydrocracking−combined acid/metal site cleavage with a unimodal distribution with >C$_{3-4}$ as major products including isomerization and unsaturated or a bimodal distribution product range in certain cases.
3. Hydrogenolysis−metal site dominated cleavage with a unimodal distribution with C$_1$ as the major product.

A schematic summarizing the product distributions for each type of mechanism is given in Fig. 1. Furthermore, these studies allowed an activity-mechanism map to be constructed for catalyst benchmarking, which should enable new, superior polyolefin depolymerization catalysts to be developed that provide specific products or distributions.

## Results

### Preparation and characterization of the catalysts

Three different types of silica-alumina were used as catalysts/supports, i.e., amorphous silica-alumina (SiO$_2$-Al$_2$O$_3$), zeolite-Y_hydrogen (Zeo-Y_H, Si/Al = 30:1) with a relatively large pore opening and zeolite socony mobil-5_hydrogen (ZSM-5_H, Si/Al = 23:1) with a relatively small pore size and similar Si/Al ratio[29–31]. Co, Ni, or Ru nanoparticles (NPs) were immobilized on the three silica-alumina materials (2.5 wt% metal loading) via a solution-based synthetic procedure (see further details in Methods) to give the main library of 12 catalysts, i.e., SiO$_2$-Al$_2$O$_3$, Co/SiO$_2$-Al$_2$O$_3$, Ni/SiO$_2$-Al$_2$O$_3$, Ru/SiO$_2$-Al$_2$O$_3$, Zeo-Y_H, Co/Zeo-Y_H, Ni/Zeo-Y_H, Ru/Zeo-Y_H, ZSM-5_H, Co/ZSM-5_H, Ni/ZSM-5_H, and Ru/ZSM-5_H. Co and Ni NPs were selected as Earth-abundant first-row transition metals with suitable characteristics for C-C bond cleavage[32–35]. Ru NPs were used as they have been shown to be particularly effective in activating C-C bonds[32,36–38]. In addition to the main library of 12 catalysts, Zeo-Y_H catalysts with various Si/Al ratios and the corresponding Ni- and Ru-modified catalysts were tested to afford a subset of 6 catalysts, i.e., Zeo-Y_H (Si/Al = 60:1) termed Zeo-Y_H [60], Zeo-Y_H (Si/Al = 80:1) termed Zeo-Y_H [80], Ni/Zeo-Y_H [60], Ni/Zeo-Y_H [80], Ru/Zeo-Y_H [60], and Ru/Zeo-Y_H [80].

All SiO$_2$-Al$_2$O$_3$-, Zeo-Y_H-, and ZSM-5_H-based catalysts were analyzed by powder-XRD, confirming the amorphous nature of SiO$_2$-Al$_2$O$_3$-based catalysts (Supplementary Fig. 1) and the crystalline nature

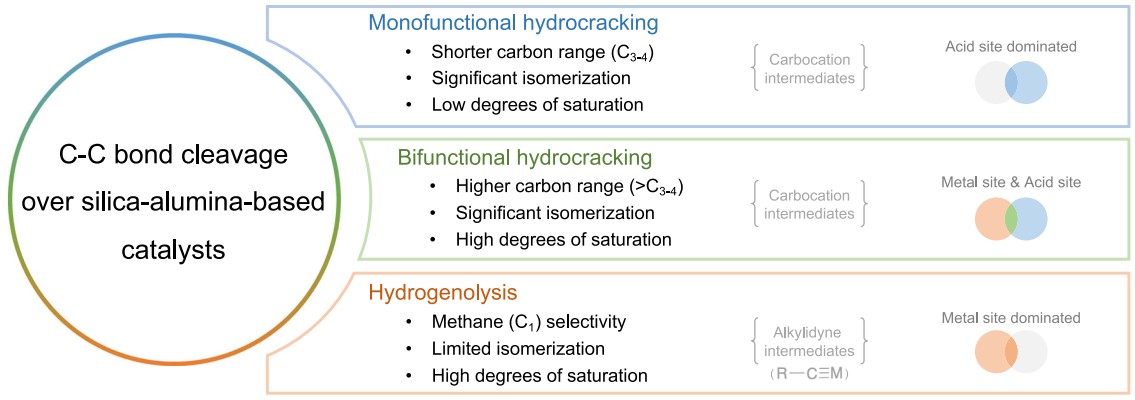

**Fig. 1 | Schematic summary of the product distributions for different C−C bond cleavage mechanisms.**

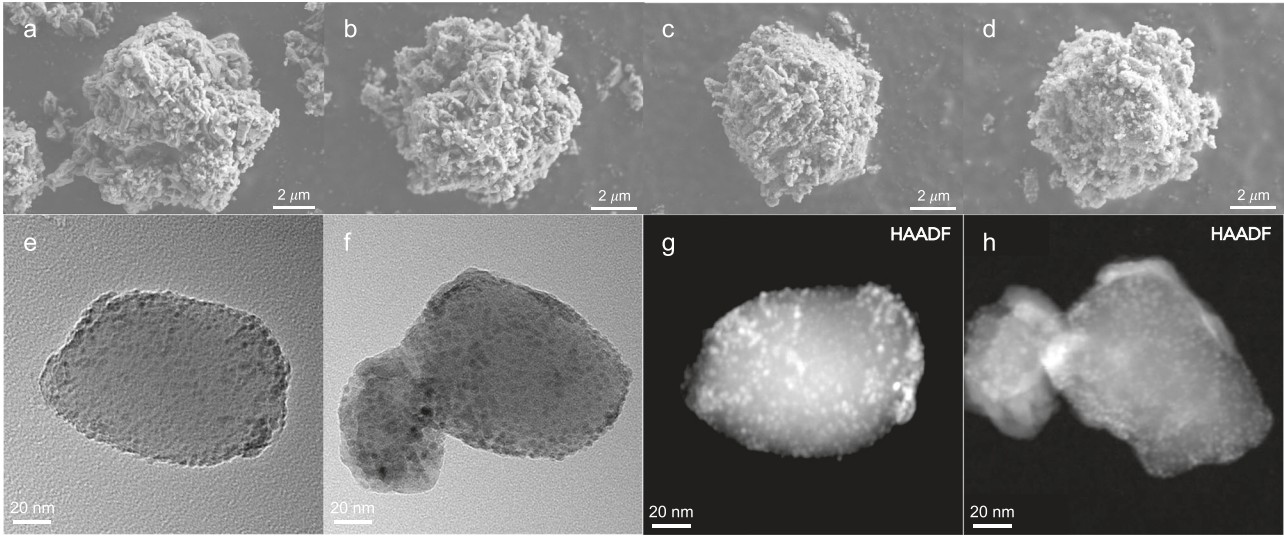

**Fig. 2 | Electron microscopy images of the ZSM-5_H-based catalysts.** SEM images (scale bar = 2 $\mu$m) of an individual **a** ZSM-5_H, **b** Co/ZSM-5, **c** Ni/ZSM-5_H, and **d** Ru/ZSM-5_H. TEM bright-field images (scale bar = 20 nm) of **e** Co/ZSM-5_H and **f** Ni/ZSM-5_H. HAADF images (scale bar = 20 nm) of **g** Co/ZSM-5_H and **h** Ni/ZSM-5_H.

of Zeo-Y_H- and ZSM-5_H-based catalysts (Supplementary Figs. 2, 3). Both the amorphous and crystalline features are well-preserved after NP immobilization to the corresponding unmodified silica-alumina materials. ZSM-5_H-based catalysts were proceeded with further characterization due to their extended applications in our studies. X-ray photoelectron spectroscopy (XPS) analysis of ZSM-5_H confirmed the surface relative atomic concentration to comprise Si (28.6%), Al (2.8%), and O (68.7%) (Supplementary Table 1). Co/ZSM-5_H, Ni/ZSM-5_H, and Ru/ZSM-5_H have surface transition metal concentrations of 9.8, 8.3, and 2.6%, respectively (Supplementary Table 1). Various ratios of metal, metal oxide, and metal hydroxide species were assigned via appropriate fitting methodology specifically for Co, Ni, and Ru at the resting state of all the catalysts (see details in Supplementary Table 2 and Supplementary Figs. 4–6)[39,40].

Scanning electron microscopy (SEM) images show that the particle size of all the ZSM-5_H-based catalysts ranges from 6 to 10 $\mu$m (Fig. 2a–d) and no significant changes compared to the support material are observed following the immobilization of Co, Ni, or Ru NPs, and the surface morphology is also maintained (cf. Fig. 2a–d with Supplementary Fig. 7). Transmission electron microscopy (TEM) images of ZSM-5_H show a smooth surface with no apparent surface structure (Supplementary Fig. 8). Well-dispersed NPs were observed on the surface of Co/ZSM-5_H and Ni/ZSM-5_H (Fig. 2e, f), and more aggregated NPs were formed on the surface of Ru/ZSM-5_H

(Supplementary Fig. 9). High-angle annular dark-field (HAADF) images also reveal the surface fine-particle structures of Co/ZSM-5_H, Ni/ZSM-5_H, and Ru/ZSM-5_H catalysts. The average particle size of the Co, Ni and Ru NPs was estimated as 3.6 ± 0.5, 3.1 ± 0.4, and 5.4 ± 0.8 nm, respectively (see further details in Supplementary Figs. 9–11). Similar particle distributions of Co and Ni NPs ensure that the further reactivity comparisons are mainly based on the types of metal modification instead of acid/metal site intimacy during their working state (hydrogen atmosphere with heat, see below)[41]. The more aggregated particle distribution, on the other hand, rationalizes the relatively lower Ru surface atomic concentration than Co and Ni via XPS analysis (2.6 vs. 9.8 and 8.3%).

### Activity studies

In the initial phase of the study *n*-hexadecane (nC$_{16}$) was used as a model substrate[23] for PE, a saturated aliphatic polymer. Each catalyst was evaluated under sufficient H$_2$ (~1.15 eq.) to quantitatively produce methane, to ensure that selectivity would not be restricted by a lack of hydrogen, and at three temperatures typical of catalytic hydrocracking reactions, i.e., 275, 325, and 375 °C[16,17,42,43]. The results from the main library catalysts are summarized in Table 2 and Supplementary Figs. 12–15 provide detailed product distributions, and Supplementary Table 3 provides the carbon balance in weight and hydrogen consumption.

**Table 2 | *n*-Hexadecane deconstructions with the 12 catalysts from the main library at 275, 325, and 375 °C**

n-hexadecane $\xrightarrow[\Delta, \text{2 hrs,}]{\text{Catalyst} \atop \text{45 bar } H_2,} C_1\text{-}isoC_{16}$

| Entry | Catalyst | Temp. (°C) | Conv. (%) | $C_{1-4}$ Yield (%)* | $C_{5-16}$ Yield (%)* |
|---|---|---|---|---|---|
| 1 | $SiO_2$-$Al_2O_3$ | 275 °C | 2.1 ± 0.2 | 0.0 | 2.1 |
| 2 | $SiO_2$-$Al_2O_3$ | 325 °C | 3.8 ± 1.0 | 0.1 | 3.7 |
| 3 | $SiO_2$-$Al_2O_3$ | 375 °C | 2.1 ± 0.2 | 0.1 | 2.0 |
| 4 | Zeo-Y_H | 275 °C | 4.5 ± 0.3 | 0.3 | 4.2 |
| 5 | Zeo-Y_H | 325 °C | 8.5 ± 0.5 | 0.7 | 7.7 |
| 6 | Zeo-Y_H | 375 °C | 26.7 ± 1.4 | 3.3 | 23.3 |
| 7 | ZSM-5_H | 275 °C | 13.7 ± 2.4 | 3.8 | 10.0 |
| 8 | ZSM-5_H | 325 °C | 91.6 ± 4.4 | 35.5 | 56.2 |
| 9 | ZSM-5_H | 375 °C | 98.0 ± 2.0 | 77.3 | 20.8 |
| 10 | Co/$SiO_2$-$Al_2O_3$ | 275 °C | 2.3 ± 0.2 | 0.1 | 2.2 |
| 11 | Co/$SiO_2$-$Al_2O_3$ | 325 °C | 2.3 ± 0.3 | 0.0 | 2.2 |
| 12 | Co/$SiO_2$-$Al_2O_3$ | 375 °C | 2.4 ± 0.1 | 0.1 | 2.2 |
| 13 | Co/Zeo-Y_H | 275 °C | 1.9 ± 0.4 | 0.0 | 1.9 |
| 14 | Co/Zeo-Y_H | 325 °C | 5.5 ± 1.2 | 0.2 | 5.2 |
| 15 | Co/Zeo-Y_H | 375 °C | 6.1 ± 1.2 | 1.2 | 4.9 |
| 16 | Co/ZSM-5_H | 275 °C | 1.9 ± 0.1 | 0.1 | 1.8 |
| 17 | Co/ZSM-5_H | 325 °C | 5.9 ± 0.1 | 1.2 | 4.7 |
| 18 | Co/ZSM-5_H | 375 °C | 49.0 ± 3.7 | 12.7 | 36.3 |
| 19 | Ni/$SiO_2$-$Al_2O_3$ | 275 °C | 2.6 ± 0.6 | 0.1 | 2.6 |
| 20 | Ni/$SiO_2$-$Al_2O_3$ | 325 °C | 3.3 ± 1.4 | 0.1 | 3.2 |
| 21 | Ni/$SiO_2$-$Al_2O_3$ | 375 °C | 3.8 ± 1.2 | 0.2 | 3.7 |
| 22 | Ni/Zeo-Y_H | 275 °C | 2.1 ± 0.6 | 0.6 | 1.5 |
| 23 | Ni/Zeo-Y_H | 325 °C | 4.4 ± 0.4 | 0.4 | 4.0 |
| 24 | Ni/Zeo-Y_H | 375 °C | 20.9 ± 1.3 | 0.4 | 20.5 |
| 25 | Ni/ZSM-5_H | 275 °C | 2.2 ± 0.2 | 0.3 | 1.9 |
| 26 | Ni/ZSM-5_H | 325 °C | 8.2 ± 1.8 | 0.5 | 7.7 |
| 27 | Ni/ZSM-5_H | 375 °C | 85.6 ± 4.1 | 28.2 | 57.3 |
| 28 | Ru/$SiO_2$-$Al_2O_3$ | 275 °C | 95.6 ± 4.0 | 41.5 | 54.2 |
| 29 | Ru/$SiO_2$-$Al_2O_3$ | 325 °C | 98.5 ± 0.3 | 75.8 | 22.6 |
| 30 | Ru/$SiO_2$-$Al_2O_3$ | 375 °C | 99.8 ± 0.2 | 96.6 | 3.3 |
| 31 | Ru/Zeo-Y_H | 275 °C | 96.0 ± 0.9 | 56.6 | 39.3 |
| 32 | Ru/Zeo-Y_H | 325 °C | 99.6 ± 0.3 | 91.5 | 8.3 |
| 33 | Ru/Zeo-Y_H | 375 °C | 99.0 ± 1.0 | 92.4 | 6.9 |
| 34 | Ru/ZSM-5_H | 275 °C | 99.4 ± 0.6 | 92.8 | 6.6 |
| 35 | Ru/ZSM-5_H | 325 °C | 98.0 ± 2.0 | 99.7 | 0.3 |
| 36 | Ru/ZSM-5_H | 375 °C | 99.8 ± 0.2 | 98.8 | 1.1 |

Reaction conditions: *n*-hexadecane (1.59 g, 7.0 mmol), catalyst (0.1 g, metal loading = 2.5 wt%), S/C ratio (substrate/catalyst weight ratio) ~16, 45 bar $H_2$, 2 h.
*All yields were calculated as the carbon yield and isomerized $C_{16}$ (*iso*$C_{16}$) are considered as products.

## Unmodified silica-alumina catalysts

Under the typical conditions used (n$C_{16}$: ~1.6 g, 7.0 mmol, 112 mmol carbon, catalyst: 0.1 g, $H_2$: 45 bar, reaction time: 2 h), $SiO_2$-$Al_2O_3$ showed limited activity even at 375 °C with a maximum conversion of 2.1 ± 0.2% (Table 2, Entries 1–3). With Zeo-Y_H the activity increases over the temperature range employed, accomplishing a conversion of 26.7 ± 1.4% at 375 °C (Table 2, Entries 4–6). ZSM-5_H achieved 13.7 ± 2.4% conversion at 275 °C (Table 2, Entry 7), and near-quantitative conversion (98.0 ± 2.0%) at 375 °C (Table 2, Entries 8, 9). Light hydrocarbon ($C_{1-5}$) products, with $C_3$ as the major products, were identified in all the reactions (Supplementary Fig. 12). Significant portions of isomerization products were detected, and limited

methane was obtained confirming a monofunctional hydrocracking mechanism. Under the reaction conditions employed, the reactivity of the silica-alumina materials follows the order: ZSM-5_H > Zeo-Y_H > $SiO_2$-$Al_2O$. The differences in activity among amorphous $SiO_2$-$Al_2O_3$, crystalline Zeo-Y_H, and crystalline ZSM-5_H at different temperatures (Table 2, Entries 1–9) reveal the significance of local confinements and the corresponding topology of a given silica-alumina material towards the activity and selectivity of the catalyst (see below)[44,45].

Control experiments were conducted in which the reaction of n$C_{16}$ was studied under a nitrogen atmosphere in the presence of the unmodified silica-alumina catalysts at 375 °C (Fig. 3a and

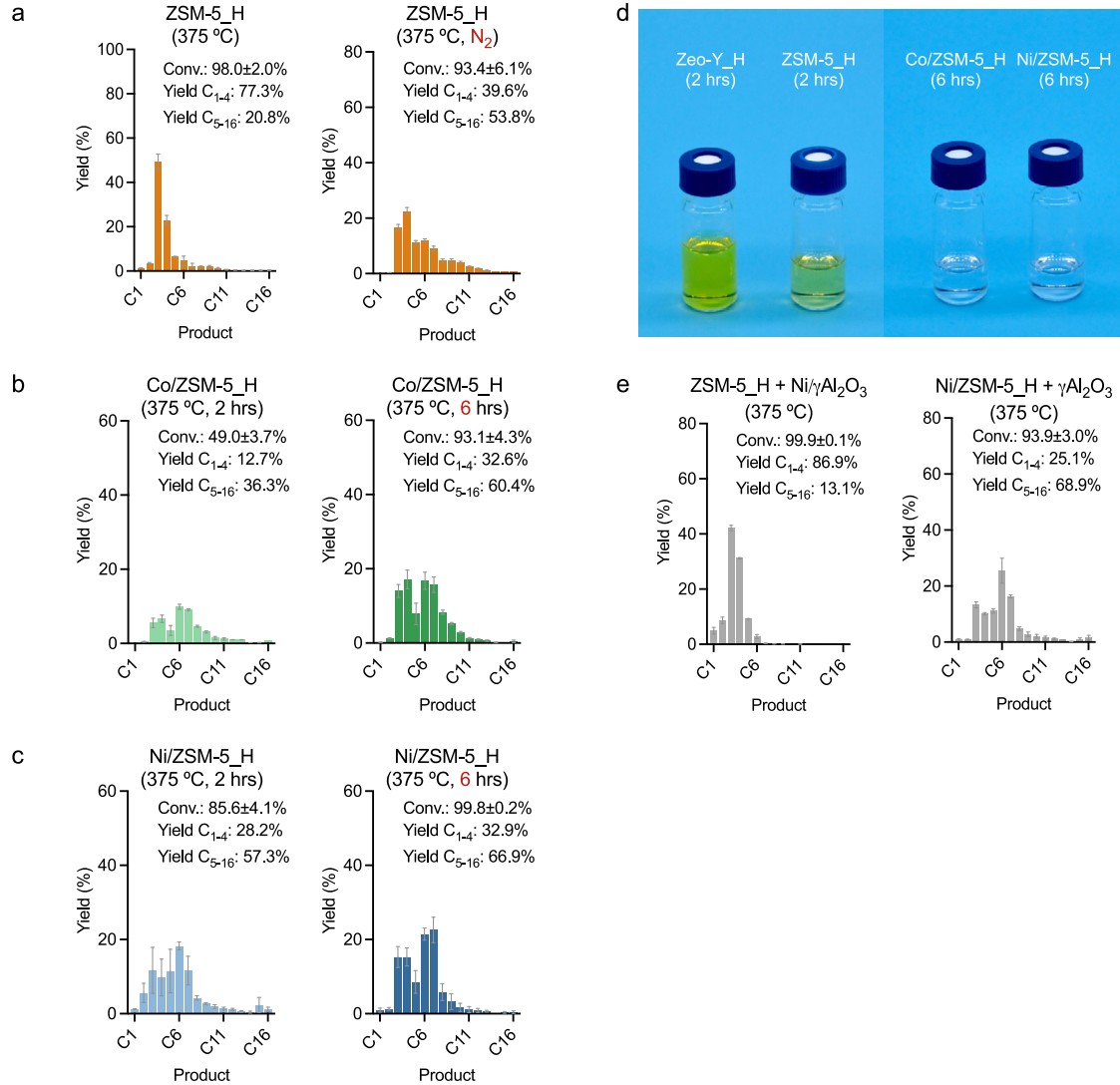

**Fig. 3 | Product distributions and appearances of liquid products after *n*-hexadecane (nC$_{16}$) deconstruction. a** nC$_{16}$ (1.59 g, 7.0 mmol) deconstruction using ZSM-5_H (0.1 g) under hydrogen and under nitrogen, pressure = 45 bar, 2 h. **b** nC$_{16}$ (1.59 g, 7.0 mmol) deconstruction using Co/ZSM-5_H (0.1 g, metal loading = 2.5 wt%), 45 bar H$_2$ for 2 and 6 h. **c** nC$_{16}$ (1.59 g, 7.0 mmol) deconstruction using Ni/ZSM-5_H (0.1 g, metal loading = 2.5 wt%), 45 bar H$_2$ for 2 and 6 h. **d** Color of the resulting liquid products from the nC$_{16}$ deconstructions using Zeo-Y_H, ZSM-5_H, Co/ZSM-5_H, and Ni/ZSM-5_H with 45 bar H$_2$ at 375 °C. **e** nC$_{16}$ (1.59 g, 7.0 mmol) deconstruction using Ni/ZSM-5_H (0.1 g, metal loading = 2.5 wt%) + γAl$_2$O$_3$ (0.1 g) and ZSM-5_H (0.1 g) + Ni/γAl$_2$O$_3$ (0.1 g, metal loading = 2.5 wt%), 45 bar H$_2$, 2 h. Error bars = standard deviation. Source data are provided as a Source Data file.

Supplementary Fig. 16). Interestingly, the nC$_{16}$ conversion in the presence of ZSM-5_H was reduced under the nitrogen atmosphere compared to the reaction under hydrogen, and a wider unimodal product distribution was obtained (cf. Fig. 3a-left and Supplementary Fig. 16-left with Fig. 3a-right and Supplementary Fig. 16-right). The major products consist of short carbon range (C$_{3-4}$) hydrocarbons with a limited amount of methane, confirming a monofunctional cracking mechanism under nitrogen. Despite the absence of an external hydrogen source when the reaction is carried out under a nitrogen atmosphere, the hydrogen atoms in the hydrocarbon substrate are able to be transferred, to afford saturated products together with the corresponding unsaturated products resulting from the hydrogen transfer[46]. The different applied atmospheres do not lead to significantly different C-C bond cleavage pathways. Nevertheless, under a hydrogen atmosphere, the reaction should be classified as hydrocracking due to the influence of the hydrogen on the kinetics and product distribution, i.e. wider vs. narrower unimodal distribution under N$_2$ or H$_2$ atmospheres[47]. Despite the unmodified silica-alumina catalysts lacking metal sites for efficient hydrogenation/

dehydrogenation steps, the hydrogen may affect the concentration and diffusion behavior of surface species[48].

### Immobilized Co and Ni NP catalysts

The Co/SiO$_2$-Al$_2$O$_3$ and Ni/SiO$_2$-Al$_2$O$_3$ catalysts only result in low conversions of nC$_{16}$, i.e., <5%, similar to that observed with the silica-alumina supports alone under the benchmark conditions at 375 °C (Table 2, cf. Entries 1–3 with 10–12 and 19–21). Co/Zeo-Y_H led to a conversion of 6.1 ± 1.2% and Ni/Zeo-Y_H led to a conversion of 20.9 ± 1.3% at 375 °C, ca. 25 and 80% of the activity of Zeo-Y_H alone (Table 2, cf. Entry 6 with 15, 24). Co/ZSM-5_H resulted in a conversion of 49.0 ± 3.7% and Ni/ZSM-5_H resulted in a conversion of 85.6 ± 4.1%, at 375 °C, ca. 50 and 90% of the activity of ZSM-5_H alone (Table 2, cf. Entry 9 with 18, 27).

A bimodal product distribution including isomerized hydrocarbons was obtained with Co/ZSM-5_H and Ni/ZSM-5_H, with both affording C$_{3-4}$ and C$_{7-8}$ hydrocarbons as major products (Fig. 3b-left, 3c-left and Supplementary Figs. 13, 14), which contrasts with a unimodal distribution dominated by C$_{3-4}$ products when ZSM-5_H was

**Table 3 | Degrees of saturation of the liquid products obtained from the deconstruction of *n*-hexadecane under hydrogen**

| Entry | Catalyst | H₂ (bar) | Time (h) | Conv. (%) | Saturated (%, δ = 0.25–2.0) | Unsaturated (%, δ = 2.0-6.0) | Aromatic (%, δ = 6.0-8.0) |
|---|---|---|---|---|---|---|---|
| 1 | ZSM-5_H | 45 | 2 | 98.0 ± 2.0 | 87.4 ± 4.3 | 7.9 ± 2.6 | 4.8 ± 1.9 |
| 2 | Co/ZSM-5_H | 45 | 2 | 49.0 ± 3.7 | 99.4 ± 0.5 | 0.5 ± 0.4 | 0.1 ± 0.1 |
| 3 | Ni/ZSM-5_H | 45 | 2 | 85.6 ± 4.1 | 97.6 ± 0.8 | 2.3 ± 0.7 | 0.1 ± 0.1 |
| 4 | Co/ZSM-5_H | 45 | 6 | 93.1 ± 4.3 | 96.2 ± 1.8 | 3.1 ± 1.6 | 0.6 ± 0.2 |
| 5 | Ni/ZSM-5_H | 45 | 6 | 99.8 ± 0.2 | 99.9 ± 0.1 | 0.1 ± 0.1 | 0.1 ± 0.1 |
| 6 | ZSM-5_H | 30 | 2 | 99.8 ± 0.1 | 86.5 ± 0.5 | 8.8 ± 0.5 | 4.7 ± 0.1 |
| 7 | ZSM-5_H | 60 | 2 | 99.9 ± 0.1 | 88.1 ± 0.2 | 7.4 ± 0.2 | 4.5 ± 0.1 |
| 8 | Ni/ZSM-5_H | 30 | 2 | 99.5 ± 0.5 | 98.6 ± 0.8 | 1.0 ± 0.7 | 0.4 ± 0.1 |
| 9 | Ni/ZSM-5_H | 60 | 2 | 94.9 ± 2.2 | 99.9 ± 0.1 | 0.1 ± 0.1 | 0.1 ± 0.1 |

Reaction conditions: *n*-hexadecane (1.59 g, 7.0 mmol), catalyst (0.1 g, metal loading = 2.5 wt%), 375 °C. Note that degrees of saturation are defined by the ratio of proton integrations in ¹H NMR spectra to indicate the adjacent carbon-carbon bonds (saturated: δ = 0.25–2.0, unsaturated: δ = 2.0–6.0, and aromatics: δ = 2.0-6.0) given the C–H and C–C bond exclusivity of hydrocarbons.

applied under the typical conditions (cf. Fig. 3a-left with 3b-left, 3c-left). The change in the product distributions may be appreciated by considering the proportion of liquid to gas ratio of the products (L/G ratio = C₅₋₁₆ yield/C₁₋₄ yield), which corresponds to 2.9 for Co/ZSM-5_H (conv. = 49.0 ± 3.7%) and 2.0 for Ni/ZSM-5_H (conv. = 85.6 ± 4.1%) compared to 0.3 for ZSM-5_H (conv. = 98.0 ± 2.0%). The bimodal product distribution and higher carbon range C₇₋₈ products obtained using Co/ZSM-5_H and Ni/ZSM-5_H implies a bifunctional pathway. A set of controls with a prolonged reaction time of 6 h in the presence of Co/ZSM-5_H and Ni/ZSM-5-H was conducted to establish whether the bimodal product distribution is due to an insufficient reaction time. Both Co/ZSM-5_H and Ni/ZSM-5_H result in near-quantitative conversion of the nC₁₆ substrate and bimodal product distributions, i.e., a L/G ratio of 1.9 for Co/ZSM-5_H (conv. = 93.1 ± 4.3%) and a L/G ratio of 2.0 for Ni/ZSM-5_H (conv. = 99.8 ± 0.2%), confirming a bifunctional hydrocracking mechanism (cf. Fig. 3a-left with 3b-right, 3c-right). The Co and Ni NPs are unable to cleave C-C bonds efficiently, and therefore the catalytic activity is mainly determined by the acid sites when the reaction is conducted above the onset temperature of the silica-alumina support, evidenced by the proportion of C₃ product for the hydrogenolysis of nC₁₆ using Co/ZSM-5_H or Ni/ZSM-5_H (Fig. 3b, c). Hence, Co/ZSM-5_H and Ni/ZSM-5_H are more active than the Zeo-Y_H-based counterparts, i.e., Co/Zeo-Y_H and Ni/Zeo-Y_H, due to the higher activity of the ZSM-5_H support, and despite having identical NP coverages. Such interplay between the immobilized NPs and surface is often referred to as the acid-metal balance and can further affect product selectivity, in these cases not specifically favoring C₄ products[49,50].

Although the catalysts composed of Co or Ni NPs immobilized on Zeo-Y_H or ZSM-5_ H are less active than the zeolite supports alone (Table 2, cf. Entries 4–9 with 13–18 and 22–27), the extent of unsaturated products (i.e., alkenes, aromatics, etc., determined using ¹H NMR spectroscopy) is lower (see further details in Table 3). This difference is because the Co and Ni NPs efficiently catalyze the hydrogenation of unsaturated bonds formed during the cracking process. For example, the degrees of saturation of the resulting liquids using ZSM-5_H under the typical conditions correspond to 87.4, 7.9, and 4.8% of saturated, unsaturated, and aromatic hydrocarbons, respectively (Table 3, Entry 1). In contrast, degrees of saturation shift to 99.4, 0.5, and 0.1% using Co/ZSM-5_H and 97.6, 2.3, and 0.1% with Ni/ZSM-5_H under the same reaction conditions (Table 3, Entries 2, 3). After a reaction time of 6 h, the degree of saturated liquid hydrocarbons increases to 96.2% for Co/ZSM-5_H and 99.9% for Ni/ZSM-5_H (Table 3, Entries 4, 5). Further control experiments using nC₁₆ as a substrate under different hydrogen pressures revealed that the degrees of saturated liquid hydrocarbons increase along the increased pressure (i.e., the amount of H₂ increases). When ZSM-5-H is applied as the catalyst, the percentage of saturated products increases from 86.5% under 30 bar of H₂ to 88.1% under 60 bar of H₂ (corresponding to ~0.80 and ~1.55 eq. required to

quantitatively produce methane, respectively, Table 3, Entries 6, 7, and Supplementary Fig. 17). With Ni/ZSM-5_H, 98.6% and 99.9% of saturated products are obtained at 30 and 60 bar of H₂, respectively (Table 3, Entries 8,9, and Supplementary Fig. 18). The higher the hydrogen pressure shows the higher the percentage of saturated products, though only to a modest extent. In contrast, the metal modification has a greater influence, significantly increasing the amount of saturated products in the final product distribution. The higher content of saturated products is a distinctive feature of the bifunctional pathway, which reduces potential coking through the suppression of coking precursors such as polyaromatic compounds[51–53]. Note that as the concentration of unsaturated products increases, the color of the solution also becomes more intensely yellow in color (Fig. 3d).

The proximity of the acid sites and metal sites was investigated by comparing the products obtained using ZSM-5_H + Ni/γAl₂O₃ (Ni-modified gamma-alumina prepared using identical procedure as Ni/ZSM-5_H) and Ni/ZSM-5_H + γAl₂O₃ as catalysts[54]. ZSM-5_H + Ni/γAl₂O₃ may be considered a low intimacy combination and Ni/ZSM-5_H + γAl₂O₃ as a high intimacy combination. The low intimacy combination resulted in near-quantitative conversion with a unimodal distribution indicative of a monofunctional pathway typical of ZSM-5_H (cf. Fig. 3a-left with 3e-left), whereas the high intimacy combination led to a conversion of 93.9 ± 0.3% with a bimodal distribution similar to the bifunctional pathway observed for Ni/ZSM-5_H (cf. Fig. 3c-left with 3e-right). These results confirm the importance of acid and metal site proximity in influencing the product distribution as reported elsewhere[41,54,55]. Note that γAl₂O₃ shows some activity (conv. = 6.8% ± 0.9) in the transformation of nC₁₆ at 375 °C, explaining the slightly higher activity of the Ni/ZSM-5_H + γAl₂O₃ combination compared to Ni/ZSM-5_H alone. Moreover, the Co and Ni metal sites appear to provide an alternative route facilitating the dissociation of intermediates away from the acid sites that would otherwise be cleaved further by them to afford higher degrees of saturated products in higher carbon ranges.

**Immobilized Ru NP catalysts**
The three Ru-modified catalysts all result in the near-quantitative conversion of nC₁₆ with high selectivity to methane, with high conversions even obtained at 275 °C (Table 2, Entries 28–36, and Supplementary Fig. 15). The high yield of methane and trace amount of isomerized products in the C₅₋₁₂ range (Table 2, Entries 28, 31) indicate that a hydrogenolysis mechanism is dominant[23,24,56]. Note that evaluating metal NPs dispersed on an inactive support material may provide insights on the onset temperature for C–C bond hydrogenolysis of a given metal. Ru NPs immobilized on carbon (Ru/C) were shown to depolymerize PE via a hydrogenolysis mechanism with an onset temperature around 220 °C, whereas Pt/C, Pd/C, and Rh/C require temperature ≥280 °C[23,24]. As Ru NPs efficiently catalyze C–C bond

hydrogenolysis, the activity and selectivity are dominated by the Ru NPs, especially where the silica-alumina support is less active, e.g., $SiO_2$-$Al_2O_3$, and below the onset temperature of the support.

Traces of isomers and chain-end initiated linear alkanes were identified at 325 °C (Supplementary Fig. 19), indicative of hydrocracking attributed to the silica-alumina supports (ZSM-5_H in this case). Therefore, the kinetics of the catalysts for methane production follows the order: Ru/ZSM-5_H (88.6%) > Ru/Zeo-Y_H (76.4%) > Ru/$SiO_2$-$Al_2O_3$ (54.5%) at 325 °C. With the exception of Ru, the influence of a metal on the selectivity of the reaction is less predictable, especially when the hydrogenolysis onset temperature is close to the temperature required by the support to initiate hydrocracking. Thus, the overhead of reaction temperature with respect to hydrogenolysis/hydrocracking onset temperature critically influences which mechanism prevails. For instance, ultra-stable Y zeolite and beta zeolite modified with Pt NPs (i.e., ≥280 °C for hydrogenolysis) behave as bifunctional hydrocracking catalysts at around 300 °C[20].

### Impact of the Si/Al ratio

Zeo-Y_H-based catalysts with different Si/Al ratios (SARs) were also evaluated in the deconstructions of nC$_{16}$ at 375 °C, see Table 4 and Supplementary Fig. 20 for the full product distributions, and Supplementary Table 4 for carbon balance in weight and hydrogen consumption. Zeo-Y_H [60] (Si/Al = 60:1) and Zeo-Y_H [80] (Si/Al = 80:1) result in conversions of 24.6 ± 1.2% and 14.9 ± 0.6% compared to the 26.7 ± 1.4% for Zeo-Y_H (Si/Al = 30:1) under the standard reaction conditions, i.e., 45 bar $H_2$, 375 °C, 2 h (Table 4, Entries 1–3). As the SAR increases the catalytic activity decreases, which may be associated with their surface acidity, and is in accordance with the literature[21]. The presence of isomerized and unsaturated products confirm that a monofunctional pathway is in operation (Supplementary Table 5, Entries 2, 3) as would be expected. The differences in activity between the different types of zeolites with comparable SARs (Si/Al = 20–30) are more considerable, i.e., Zeo-Y_H results in a conversion of 26.7 ± 1.4% whereas ZSM-5_H results in a conversion of 98.0 ± 2.0% under the standard reaction conditions (cf. Table 2, Entries 6, 9 with Table 4, Entries 2, 3). Such difference also reveals that the confinements originating from the zeolite topology play a more dominant role than the SAR[57]. Furthermore, Zeo-Y_H shows selectivity toward higher carbon range products due to the larger pore sizes whereas ZSM-5_H favors the C$_{1-4}$ hydrocarbons which may be attributed to smaller pores, typically referred as shape selectivity and correlated to the zeolite topology[30,58,59].

Ni/Zeo-Y_H [60] and Ni/Zeo-Y_H [80] result in conversions of 5.8 ± 0.5% and 4.9 ± 0.7% of nC$_{16}$ under 45 bar $H_2$ at 375 °C reacting for 2 hs compared to the conversion of 20.9 ± 1.3% via Ni/Zeo-Y_H (Table 4, Entries 4–6). Similar trends were observed to that of the main library catalysts, i.e., the activity decreases following the immobilization of Ni NPs of Ni/Zeo-Y_H [60] and Ni/Zeo-Y_H [80] (Table 4, cf. Entries 2, 3 with 5, 6). The activity of the catalysts follows the order: Ni/Zeo-Y_H > Ni/Zeo-Y_H [60] > Ni/Zeo-Y_H [80], the same order as the unmodified supports, and confirms that C-C bond cleavage principally depends on the acid sites of the silica-alumina support at 375 °C. High degrees of saturated products obtained using Ni/Zeo-Y_H [60] and Ni/Zeo-Y_H [80] confirm a bifunctional pathway (Supplementary Table 5, Entries 5, 6). Both Ru/Zeo-Y_H [60] and Ru/Zeo-Y_H [80] led to quantitative conversion as observed for Ru/Zeo-Y_H at 375 °C (Table 4, Entries 7–9), with methane as the dominant product, confirming the hydrogenolysis pathway is in operation (Supplementary Fig. 20).

### PE depolymerization

The catalysts employing ZSM-5_H as the support material are the most active and were further used to depolymerize PE (Mw ~4000 g/mol, Sigma-Aldrich). ZSM-5_H results in a conversion of 95.8% (PE: ~1.6 g, ~112 mmol of carbon, catalyst: 0.2 g, $H_2$: 45 bar, reaction time: 2 h, solvent-free) at 375 °C. The product distribution is similar to that obtained with nC$_{16}$, i.e., with C$_{3-4}$ hydrocarbons as the major products (Fig. 4a) and unsaturated liquid products, confirming a monofunctional hydrocracking mechanism. Under identical reaction conditions, quantitative depolymerization of PE to methane was observed in the presence of Ru/ZSM-5_H, clearly demonstrating a hydrogenolysis mechanism (Fig. 4b).

Low conversions of 13.3 and 27.1% were obtained at 375 °C in the presence of Co/ZSM-5_H and Ni/ZSM-5_H, respectively (Fig. 4c, d). Prolonging the reaction time to 16 h increases the conversion of PE to 66.7 ± 4.4% for Co/ZSM-5_H and 87.1 ± 7.5% for Ni/ZSM-5_H (Fig. 4e, f). With the prolonged reaction time of 16 h, unimodal product distributions were observed with C$_{3-4}$ hydrocarbons being the major products using Co/ZSM-5_H and Ni/ZSM-5_H under solvent-free conditions. Inherent mass- and heat-transfer limitations of the solvent-free reaction involving a polymeric starting material[60–62], could be mitigated by employing a solvent[63,64]. Hence, nC$_{16}$ was applied as a reactive solvent (note that the total carbon molar number was maintained, i.e., nC$_{16}$ + PE = ~1.4 + 0.2 g corresponding to ~112 mmol of carbon, catalyst = 0.1 g). Significant coking was not observed presumably due to the short reaction time and the application of hydrogen (Tables 2, 4),

**Table 4 | n-Hexadecane deconstruction with the additional Zeo-Y_H-based catalysts with varying SARs**

n-hexadecane $\xrightarrow[\text{45 bar } H_2,\ \text{375 °C, 2 hrs,}]{\text{Catalyst}}$ C$_1$- isoC$_{16}$

| Entry | Catalyst | Conv. (%) | C$_{1-4}$ Yield (%)* | C$_{5-16}$ Yield (%)* |
|---|---|---|---|---|
| 1 | Zeo-Y_H | 26.7 ± 1.4 | 3.3 | 23.3 |
| 2 | Zeo-Y_H [60] | 24.6 ± 1.2 | 3.8 | 20.8 |
| 3 | Zeo-Y_H [80] | 14.9 ± 0.6 | 2.0 | 12.8 |
| 4 | Ni/Zeo-Y_H | 20.9 ± 1.3 | 0.4 | 20.5 |
| 5 | Ni/Zeo-Y_H [60] | 5.8 ± 0.5 | 0.9 | 4.9 |
| 6 | Ni/Zeo-Y_H [80] | 4.9 ± 0.7 | 0.6 | 4.3 |
| 7 | Ru/Zeo-Y_H | 99.0 ± 1.0 | 92.4 | 6.9 |
| 8 | Ru/Zeo-Y_H [60] | 99.0 ± 1.0 | 98.0 | 2.0 |
| 9 | Ru/Zeo-Y_H [80] | 99.0 ± 1.0 | 97.2 | 2.8 |

Reaction conditions: n-hexadecane (1.59 g, 7.0 mmol), catalyst (0.1 g, metal loading = 2.5 wt%), S/C ratio (substrate/catalyst weight ratio) ~16, 45 bar $H_2$, 375 °C, 2 h.
*All yields were calculated as the carbon yield and isomerized C$_{16}$ (isoC$_{16}$) are considered as products.

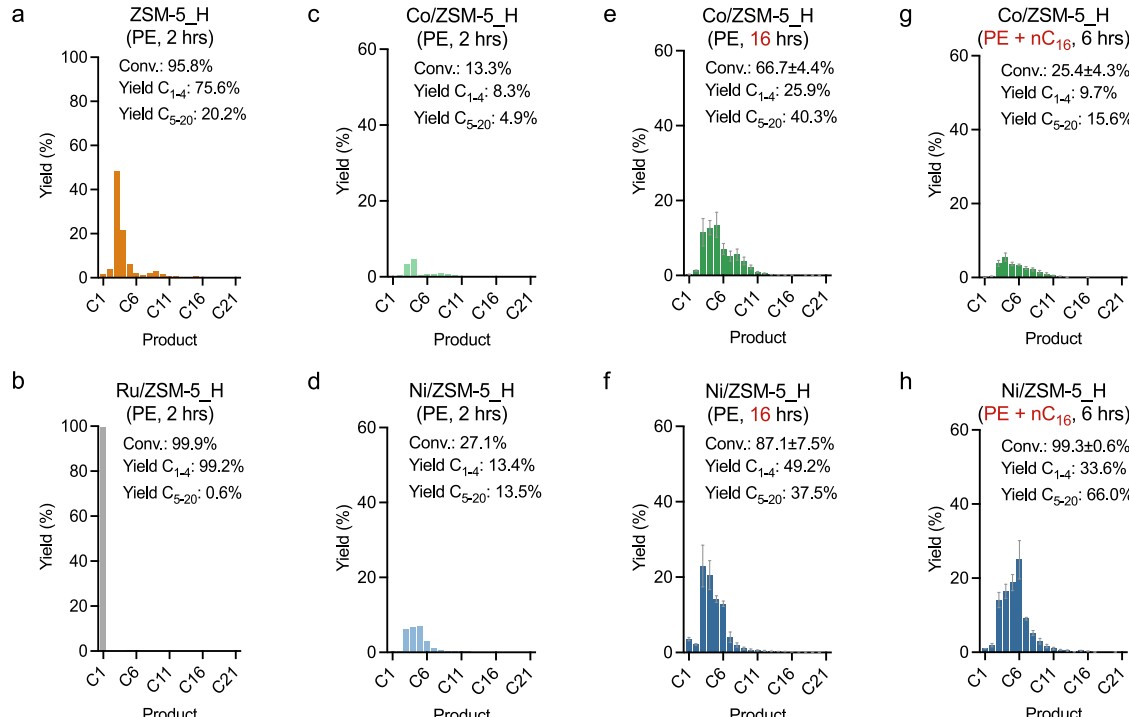

**Fig. 4 | Polyethylene (PE) depolymerization under solvent-free and solvent-assisted conditions. a** PE (1.59 g, ~112 mmol of carbon) depolymerization under solvent-free condition: catalyst (0.2 g, metal loading = 2.5 wt%), 45 bar $H_2$, 375 °C, 2 h via ZSM-5_H, **b** Ru/ZSM-5_H, **c** Co/ZSM-5_H, **d** Ni/ZSM-5_H, **e** Co/ZSM-5_H, reaction time = 16 h, **f** Ni/ZSM-5_H, reaction time = 16 h. **g** PE (0.2 g, ~14 mmol of carbon) depolymerization under the solvent-assisted condition: $nC_{16}$ (1.4 g, ~98 mmol based on carbon, as a reactive solvent), catalyst (0.1 g, metal loading = 2.5 wt%), 45 bar $H_2$, 375 °C, 6 h via Co/ZSM-5_H and **h** Ni/ZSM-5_H. Error bars = standard deviation. Source data are provided as a Source Data file.

evidenced by the similar curves obtained from thermogravimetric analysis (TGA) of fresh and used catalysts (Supplementary Fig. 21). No apparent differences in TGA curves above 500 °C were observed, indicating a well-closed carbon balance by the gaseous and liquid products due to limited solid residual formation during a reaction. The use of $nC_{16}$ a solvent is also expected to prevent coke formations and to preserve catalytic performance for the more challenging polymer substrates over longer-term operation[65,66], particularly in the cases via highly acidic catalysts[67–69]. Nonetheless, regeneration processes for coke removal could be applied if required[70,71].

Co/ZSM-5_H shows the conversion of 25.4 ± 4.3%, whereas Ni/ZSM-5_H results in a near-quantitative conversion of 99.3 ± 0.6% at 375 °C with a reaction time of 6 h (Fig. 4g, h). No significant differences in the product distribution or selectivity were observed with Co/ZSM-5_H under solvent-free or solvent-assisted depolymerization conditions, presumably due to the general low activity of the applied catalyst (cf. Fig. 4e with 4g). However, the selectivity shifts toward longer hydrocarbons with Ni/ZSM-5_H under solvent-assisted conditions, resulting in $C_{5–6}$ hydrocarbons as the main products (cf. Fig. 4f with 4h). The intrinsic activity of a given catalyst, therefore, has more significant effects on transfer issues toward selectivity and should be considered a primary factor. Despite the shift toward a higher carbon range when employing the solvent-assisted condition, it is worth noting that the resulting unimodal distribution with $C_{5–6}$ as the major products after PE depolymerization in the presence of Ni/ZSM-5_H differs from the bimodal distribution obtained for $nC_{16}$ (cf. Fig. 3c-right with 4h). Co- and Ni-modified catalysts afford predominantly saturated products following the PE depolymerizations, indicating a bifunctional hydrocracking mechanism for PE depolymerization in the presence of Co/ZSM-5_H and Ni/ZSM-5_H under solvent-assisted conditions. These, however, revealed a more sensitive nature of the bifunctional pathway upon transfer issues than the two other pathways. Recent studies also demonstrated that the migration of intermediates

between the acid and metal sites affects reactivity as well as selectivity, and rationalizes the sensitive nature of a bifunctional pathway compared to the monofunctional and hydrogenolysis pathways which have more localized active sites[53,54].

## Activity-mechanism map

The assignment of the principal mechanism for each catalyst is based on the product distribution, i.e., carbon range, isomerization, and unsaturated products. The catalysts were plotted on an activity-mechanism map in a polar coordinate system (Fig. 5a) for the reactions of $nC_{16}$ deconstructions under 45 bar $H_2$ at 375 °C with a reaction time of 2 h (i.e., the benchmark conditions). Differences in activity (r) and mechanistic pathway (θ) are indicated by the map (see further details in Methods), allowing rapid benchmarking of new catalysts and enabling the prediction of specific product distributions for depolymerizations.

The types of metal modifications strongly influence the principal cleavage mechanism (Fig. 5b–e). All unmodified silica-alumina supports function via a similar mechanism as confirmed by the similar product distributions (cf. Fig. 5a with Supplementary Fig. 12). Immobilization of Co and Ni NPs on the silica-alumina supports switches the mechanism to bifunctional hydrocracking with the Ni-modified catalysts being more active than the Co-analogs (cf. Fig. 5c with 5d, e). The catalysts containing immobilized Ru NPs operate predominantly via hydrogenolysis as methane is the major product (cf. Fig. 5a with Supplementary Fig. 15). The types of silica-alumina support affect the rate of reaction considerably via the acidic and confinement of the support structure but not the principal mechanism (cf. Fig. 5b–e with Supplementary Fig. 22)[72–75]. The Si/Al ratio influences the catalytic activity but to a lesser degree than the type of metal NPs used and the topology of the silica-alumina support (Supplementary Fig. 23). In contrast, the applied reaction temperature has shown no apparent patterns for

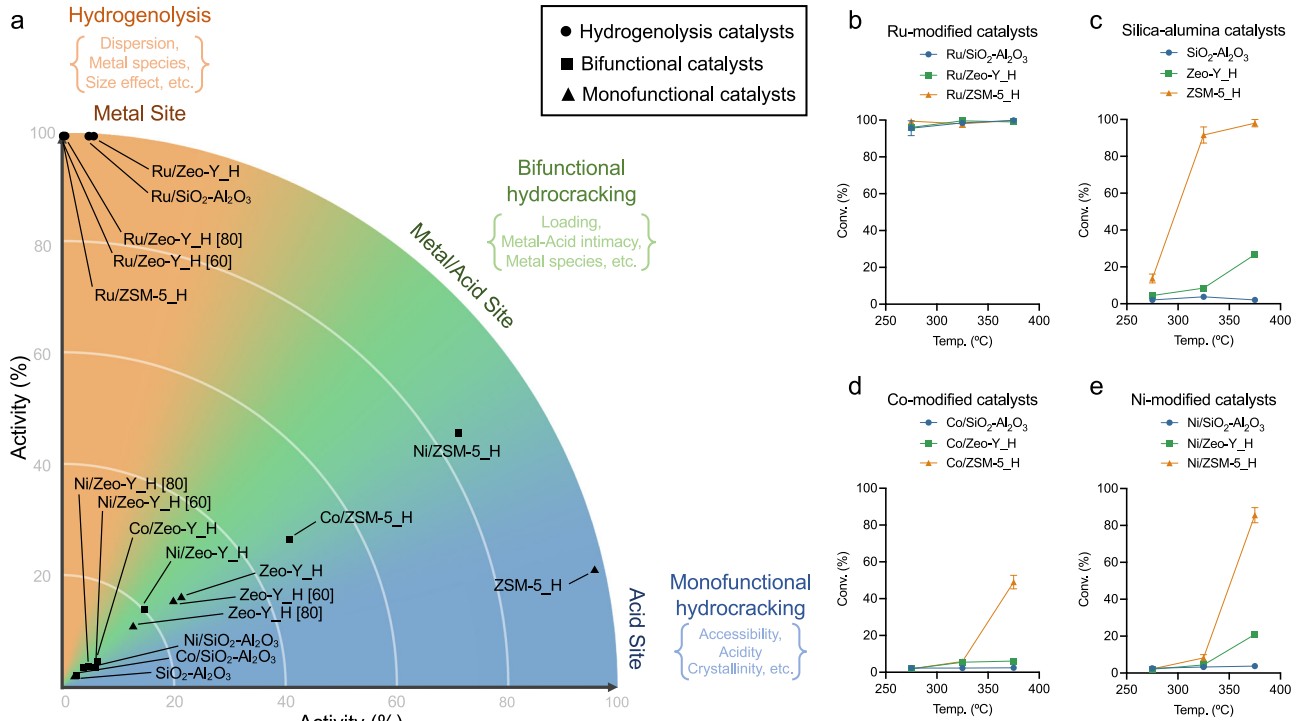

**Fig. 5 | Activity-mechanism map and catalytic *n*-hexadecane (nC₁₆) decompositions via types of metal modifications. a** Activity-mechanism map with the 18 tested catalysts positioned according to their activity and mechanism (note that the activity is denoted by the nC₁₆ conversion at the benchmark condition: 45 bar H₂, 375 °C, 2 h). C-C bond cleavage is dominated by hydrogenolysis (orange area), bifunctional hydrocracking (green area), or monofunctional hydrocracking (blue area). **b** nC₁₆ (1.59 g) deconstructions with 45 bar H₂, at 275, 325, and 375 °C, 2 h, via

Ru-modified catalysts (0.1 g, metal loading = 2.5 wt%): Ru/SiO₂-Al₂O₃, Ru/Zeo-Y_H, and Ru/ZSM-5_H, **c** unmodified silica-alumina catalysts (0.1 g): SiO₂-Al₂O₃, Zeo-Y_H, and ZSM-5_H, **d** Co-modified catalysts (0.1 g, metal loading = 2.5 wt%): Co/SiO₂-Al₂O₃, Co/Zeo-Y_H, and Co/ZSM-5_H, and **e** Ni-modified catalysts (0.1 g, metal loading = 2.5 wt%): Ni/SiO₂-Al₂O₃, Ni/Zeo-Y_H, and Ni/ZSM-5_H. Error bars = standard deviation. Source data are provided as a Source Data file.

predicting activity and prevailing mechanism (Supplementary Fig. 24), unless the onset temperatures of the catalytic components are known.

## Discussion

The depolymerization of polyolefins is an attractive way to recycle/reuse the plastic waste in comparison to mechanical recycling to afford high-quality products[76–78] and has a greater impact on sustainability than landfilling[79,80]. Despite the current cost of hydrogen, which is predicted to decrease[81,82], the ability to fine-tune the reaction mechanism and hence the product distribution is highly valuable. Moreover, as a direct consequence of the reductive environment, the process is expected to be cleaner and result in less coke formation than alternative methods such as incineration and pyrolysis[83–85].

In order to transfer the process from the laboratory scale to the industrial arena, new catalysts with superior activity and selectivity to the current range are required. To this end, we classified depolymerization mechanisms according to the product distribution, which includes the carbon range, isomerization, and degrees of saturation. Based on the study of 18 catalysts, an activity-mechanism map was constructed according to the defined benchmark conditions employing nC₁₆ as a model substrate. The activity-mechanism map serves three major functions. First, it can be used to classify catalysts that afford a specific product distribution. Second, new catalysts can be plotted on the map to allow their activity and mechanism to be gauged relative to the known catalysts. Third, the optimal components, i.e., support, metal type, etc., to provide a specific product distribution can be ascertained from the map. We expect this benchmarking strategy accelerates the development of new catalysts for both plastic-to-fuel and plastic-to-chemical scenarios.

## Methods

### General

All reactions with nC₁₆ deconstructions and certain reactions with PE depolymerizations were performed at least 3 times, and the statistics of average conversion and standard deviation were calculated. *n*-Hexadecane and polyethylene (PE, Mw ~4000, Mn ~1700) were purchased from Sigma-Aldrich. *n*-Dodecane was acquired from Abcr. CoCl₂, NiCl₂·6H₂O, and RuCl₃·3H₂O were purchased from ChemPur, Abcr, and Precious Metal Online, respectively. All metal salts were stored in desiccators. Amorphous silica-alumina (catalyst support, grade 135), zeolite-Y_hydrogen (Si/Al = 30:1, 780 m²/g; Si/Al = 60:1, 720 m²/g; Si/Al = 80:1, 780 m²/g), and ZSM-5_ammonium (Si/Al = 23:1, 425 m²/g) were obtained from Sigma-Aldrich, Abcr, and Zeolyst, respectively. ZSM-5_ammonium (2.0 g) was calcined at 550 °C under 300 mL/min dry air for 60 min to transform into ZSM-5_hydrogen[86]. Gamma-alumina (γAl₂O₃ = 97% min., 185 m²/g) was purchased from Strem.

### Typical synthesis of metal-modified catalyst

The support material (1.50 g) and a metal salt (CoCl₂, 82.7 mg, 0.64 mmol; NiCl₂·6H₂O, 151.8 mg, 0.64 mmol; RuCl₃·3H₂O, 97.1 mg, 0.37 mmol) were placed in a round bottle flask (100 mL) with a magnetic stir bar and then dispersed in deionized (DI) water (36 mL). The suspension was stirred at 500 rpm at 35 °C for 1 h. NaBH₄ (70.1 mg, 1.85 mmol for RuCl₃·3H₂O; 120.8 mg, 3.19 mmol for CoCl₂ and NiCl₂·6H₂O; 5 eq. to the applied metal) was dissolved in DI water (36 mL) and added to the suspension in one portion. The suspension was stirred at 800 rpm for 1 h. The resulting solid was vacuum filtered and washed with DI water (3 × 25 mL). The solid was then dried at 100 °C in an oven for 18 h. Various batches of the catalyst were prepared, characterized, and evaluated, and no significant differences were observed.

## Typical reaction of n-hexadecane deconstruction

n-Hexadecane (1.59 g, 7.0 mmol; 112 mmol based on carbon) and the catalyst (0.10 g) were added to a glass vial (20 mL) with a glass magnetic stir bar, and the vial was then placed into an autoclave (75 mL, Parr Instrument). The autoclave was purged three times with $H_2$ and then pressurized to 45 bar and sealed. The pressurized autoclave was placed in a heating block at the desired temperature (275–375 °C) and stirred (350 rpm) for the given reaction time (2–6 h). After the reaction, the autoclave was cooled to room temperature in a water bath. The gaseous products were transferred into a gas-sampling bag (1 L) and injected into a specialized gas-sampling gas chromatography-flame ionization detector (GC-FID) for analysis. For GC-mass spectrometry (GC-MS) analysis of the liquid products, n-dodecane (0.085 g, 0.5 mmol) was added to the vial as an internal standard and diethyl ether was used as the solvent if the resulting liquid was insufficient for GC sample preparation. In certain cases, dichloromethane was used to wash (8 mL × 3) the catalyst, and the catalyst was then vacuum dried overnight before conducting the thermogravimetric analysis (TGA).

## Typical reaction of polyethylene depolymerization

PE (1.59 g, Mw ~4000, Sigma-Aldrich; ~112 mmol of carbon) and the catalyst (0.20 g) were added to a glass vial (20 mL) with a glass-coated magnetic stir bar, and the vial was then placed into an autoclave (75 mL, Parr Instrument). The autoclave was purged three times with $H_2$ and then pressurized to 45 bar and sealed. The pressurized autoclave was placed in a heating block at the desired temperature (275–375 °C) and stirred (500 rpm) for the given reaction time (2–16 h). After the reaction, the autoclave was cooled to room temperature in a water bath. The gaseous products were transferred into a gas-sampling bag (1 L) and injected into a specialized gas-sampling gas chromatography-flame ionization detector (GC-FID) for analysis. For GC-mass spectrometry (GC-MS) analysis of the liquid products, n-dodecane (0.085 g, 0.5 mmol) was added to the vial as an internal standard and diethyl ether was used as the solvent if the resulting liquid was insufficient (<0.5 mL) for GC sample preparation.

## Instruments

Analysis of the liquid products was performed using an Agilent 7890B Gas Chromatograph coupled with Agilent 7000C MS triple quad detector with He as a carrier gas, Agilent HP-5ms ultra inert capillary column, with a ramping rate of 7.5 °C/min from 40 to 197.5 °C (for samples of $nC_{16}$ conversion) or from 40 to 295 °C (for samples of PE conversions). Analysis of gaseous products was performed using an Agilent 7890B Gas Chromatograph with $N_2$ as a carrier gas, Agilent PoraPLOT Q capillary column, with a ramping rate of 10 °C/min from 35 to 175 °C. The corresponding retention times of $C_{1-5}$ hydrocarbons was identified by a reference gas mixture obtained from Carbagas comprising $H_2$, CO, $CO_2$, $CH_4$, $C_2H_4$, $C_2H_6$, $C_3H_6$, $C_3H_8$, and $nC_4H_{10}$ in $N_2$. FID area signals were calibrated using reference methane, ethane, propane, and n-butane gas bottle purchased from Sigma-Aldrich.

$^1$H NMR spectra were recorded on a Bruker 400 MHz instrument. Powder X-ray powder diffraction (Powder-XRD) patterns were acquired by Bruker D8 Discover. X-ray photoelectron spectroscopy (XPS) measurements were obtained on a Kratos Axis Supra. Scanning electron microscopy (SEM) images were recorded on a Carl Zeiss Gemini 300 microscope. Transmission electron microscopy (TEM) images and scanning transmission electron microscopy (STEM) mappings were obtained on the FEI Talos microscope and analyzed under 200 keV acceleration energy. Thermogravimetric analysis (TGA) was conducted by Mettler Toledo TGA/DSC 3+ with a ramping rate of 5 °C/min from 35 to 900 °C and a flow rate of 20 mL/min under air.

## Data analysis and quantification

Conversion of $nC_{16}$ and quantification of the $C_{5-16}$ products was estimated from the GC-FID signal ratio with $nC_{12}$ as the internal standard with documented approaches of effective carbon number (ECN) calculations for correlations among the resulted hydrocarbon products[87]. Quantification of the $C_{1-4}$ fraction was determined from the GC-FID signal ratio compared to signals with 100% reference gas injection.

$$C_{k(k=5-16)} \text{ Yield } (\%) = \frac{n_{(C_{k,\text{act.}})}}{n_{(C_{k,\text{theo.max}})}} * 100$$

$$= \frac{\text{Coef.}_{(ECN)} * \frac{\text{Area}_{(c_{k,\text{act.}})}}{\text{Area}_{(nC_{12,\text{int.std.}})}} * n_{(nC_{12,\text{int.std.}})}}{n_{(C_{k,\text{theo.max}})}} * 100 \tag{1}$$

$$C_{l(l=1-4)} \text{ Yield } (\%) = \frac{n_{(C_{l,\text{act.}})}}{n_{(C_{l,\text{theo.max}})}} * 100$$

$$= \frac{\frac{\text{Area}_{(C_{l,\text{act.}})}}{\text{Area}_{(c_{l,\text{ref.gas}})}} * \frac{RT}{P_f V_{(\text{autoclave})}}}{n_{(C_{l,\text{theo.max}})}} * 100 \tag{2}$$

Conversion of PE was calculated using the following equation and quantification of the products was determined from the GC-FID signal ratio using $nC_{12}$ as an internal standard:

$$\text{Conversion } (\%) = \frac{m_{i(PE)} - m_{f(\text{unconverted PE})}}{m_{i(PE)}} * 100 \tag{3}$$

## Activity-mechanism map

The activity-mechanism map is constructed on a two-dimensional polar coordinate system composed of activity as the radius (r) and mechanism as the theta ($\theta$). The value for "activity" is denoted by the $nC_{16}$ conversion at the selected benchmark conditions (45 bar $H_2$, 375 °C, 2 h as the current case). The benchmark conditions would be further adapted (preferably lower applied temperature) in the case of a new catalyst that outperforms all the tested catalysts by relocating the maximum activity ($r = 100\%$). The value for "mechanism" is denoted by the similarity compared to a perfect mechanism with product selectivity as the major feature:

0° (perfect monofunctional hydrocracking): $C_{3-4}$ products only.
45° (perfect bifunctional hydrocracking): >$C_{3-4}$ products only.
90° (perfect hydrolysis): $C_1$ product only.

## Data availability

The data that support the figures and findings of this work are provided in the Supplementary Information and in the Source Data files. Source data are provided with this paper.

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

## Acknowledgements

This work is supported by the EPFL (Switzerland) and created as part of NCCR Catalysis (grant number 180544), a National Centre of

Competence in Research funded by the Swiss National Science Foundation. We thank prof. Holger Frauenrath, Julian Bleich, Matthieu Wendling, Yann Lavanchy, and Yevhen Hryshunin for technical support.

## Author contributions

W.-T.L., A.v.M., F.D.B., and P.J.D. contributed to the design of the experiments and data analysis. W.-T.L. and J.R.C. performed the experiments and M.D.M. performed the XPS analysis; W.-T.L. and P.J.D. wrote the manuscript, and all the authors discussed, commented on, and proofread the manuscript.

## Competing interests

W.-T.L., A.v.M., F.D.B., and P.J.D. are founders of a spin-off company. The remaining authors declare no competing interests.
