## [Peer Review File · Nature Communications]

Title: Polyolefin depolymerization over silica-alumina-based catalysts: mechanistic classifications and benchmarkingREVIEWER COMMENTS

Reviewer #1 (Remarks to the Author):

Lee et al. reported the metal/zeolite catalyzed C-C cleavage for the cracking of PE and hexadecane model molecule. It was found the reaction conditions, metals, and zeolites are all crucial factors for obtaining the gaseous and liquid products. These results might provide some information on the catalytic performance-catalyst composition relationships, as called activity-mechanism map in the manuscript. However, some important results and deep discussion are missing in the current version. Further works on the following issues are required for a revision.

(1) For the aluminosilicate catalysts, the Si/Al ratios have been found crucial for the catalytic activity and the product selectivity, as described in the Ref. 21. The different zeolite supported metal catalyst, the Si/Al ratios should be varied to identify the control factor for the performance.

(2) The ZSM-5 supported Co and Ni catalysts were superior compared with the Y supported Co/Ni catalyst, which need an explanation. In the previous tests such as the results in Ref. 21, the ZSM-5 favors the formation of light molecules (e.g. C1-C4), while the Y zeolite with comparable Si/Al ratios could form liquid products. The Si/Al ratios and topological structure should be discussed separately.

(3) The coke formation during the catalysis should be mentioned. As known, the zeolite with strong acidity easily caused the coking in the PE depolymerization at high temperature. This feature would reduce the recyclability of the zeolite-based catalysts for PE depolymerization without further coke-removal treatment.

(4) How about the reaction channels to form liquid and gaseous products? In the Pt and zeolite tandem catalysis at temperatures lower than 300 C, the hydrogenolysis is regarded for the C-C cleavage, while the zeolite catalyzes the secondary cracking reactions to optimize the product selectivity. The Co and Ni would have lower activity than Pt for the hydrogenolysis, and zeolite catalyzed cracking might be dominant for PE cracking at high temperature (e.g. 375 C). The functions of Co/Ni metals and zeolite should be described in the revised manuscript.

(5) Carbon balance values are suggested to be included in Figure 2 and Table 2.

Reviewer #2 (Remarks to the Author):

This paper studies hydrogenolysis of n-hexadecane in a batch reactor as a model polyolefin feed. The key learning from this paper is shown in Figure 4 where the author plot the activity of several different catalysts. The authors have tested several catalysts and combine it in this figure. Several recent studies have been done on hydrogenolysis of plastics into an oil mixture including the work of Y Roman and D

Vlachos. I do not see any new scientific learnings about the catalysis from this study compared to those other studies and am concerned about the novelty of this work. Hydrogenolysis is going to be an expensive reaction to do because of the high hydrogen requirements and the low value of the products that are produced (mainly fuels or lubricants). I recommend the authors report their hydrogen consumption. I am not sure why they did not start with an alkene rather than an alkane as pyrolysis of polyolefins produces alkenes as the primary product. There has been lots of work on conversion of polyolefins over zeolite catalysts. I am however not sure what is learned from this study that is not already known in the literature.

Here are a few other comments:

1. What is the formula for yield?
2. Why are the error bars on Ni/Co yields are much larger compared to the yield using ZSM-5 in Figure 2? Some of the error bars are almost as big as the yield of some products.
3. More evidence/discussion is required to classify the reaction mechanism. I do not think classifying by product distribution is sufficient.
4. Discussion about the confinement effects should be added.
5. Proximity of acid sites and metal sites should also be included. I do not expect an extensive analysis but at least some demonstration is necessary.

Reviewer #3 (Remarks to the Author):

Dyson et al report the conversion of a low molecular weight polyethylene into liquid and gaseous hydrocarbon products with a series of supported metal catalysts on acid supports (amorphous silica-alumina and zeolites). In general, the manuscript is well written and organized, the concept is very interesting and firmly examined. In summary, the work presented here merits publication in Nat. Commun. and will certainly be of interest to the readership in this journal. However, there are several specific questions for this manuscript and they are listed below.

Where is the hydrogen coming from in the metal-free reactions with N₂? Does the degree of unsaturation from the final product account for how much hydrogen would need to be consumed to convert the reactant to C₁ – C₈s?

Would be interesting if the authors varied H₂ pressure and what effect that would have on unsaturated product formation, since they claim a benefit of the bifunctional pathway is suppressing coke precursors.

What about the bifunctional catalysts make C₃ and C₅/C₆ the major peaks in a bimodal distribution? Why is C₄ an unlikely product here? Having a scheme showcasing these reactions and discussing why

certain products are/are not favorable would make this argument easier to follow. The authors map the catalysts according to their hydrogenolysis vs. hydrocracking activity, and so being able to visualize specific mechanisms responsible for each transformation and how each catalyst falls in line with a scaling combination of both would make more sense with the product distribution given.

Reviewer #1 (Remarks to the Author):

Lee et al. reported the metal/zeolite catalyzed C-C cleavage for the cracking of PE and hexadecane model molecule. It was found the reaction conditions, metals, and zeolites are all crucial factors for obtaining the gaseous and liquid products. These results might provide some information on the catalytic performance-catalyst composition relationships, as called activity-mechanism map in the manuscript. However, some important results and deep discussion are missing in the current version. Further works on the following issues are required for a revision.

(1) For the aluminosilicate catalysts, the Si/Al ratios have been found crucial for the catalytic activity and the product selectivity, as described in the Ref. 21. The different zeolite supported metal catalyst, the Si/Al ratios should be varied to identify the control factor for the performance.

Six additional catalysts were prepared in which the Si/Al ratio of the zeolite support, i.e., Zeo-Y_H [60] (Si/Al = 60:1) and Zeo-Y_H [80] (Si/Al = 80:1), obtained from the same supplier as the Zeo-Y_H (Si/Al = 30:1). The 6 catalysts were evaluated in the benchmark reaction, and the outcome of these studies together with additional discussions and citations to the literature have been included in the revised manuscript and Supplementary Information:

In addition to the main library of 12 catalysts, Zeo-Y_H catalysts with various Si/Al ratios and the corresponding Ni- and Ru-modified catalysts were tested to afford a subset of 6 catalysts, i.e., Zeo-Y_H (Si/Al = 60:1) termed Zeo-Y_H [60], Zeo-Y_H (Si/Al = 80:1) termed Zeo-Y_H [80], Ni/Zeo-Y_H [60], Ni/Zeo-Y_H [80], Ru/Zeo-Y_H [60] and Ru/Zeo-Y_H [80].

Impact of the Si/Al ratio

Zeo-Y_H-based catalysts with different Si/Al ratios (SARs) were also evaluated in the deconstructions of nC_{16} at 375 °C, see Table 4 with Supplementary Fig. 20 for the full product distributions. Zeo-Y_H [60] (Si/Al = 60:1) and Zeo-Y_H [80] (Si/Al = 80:1) result in conversions of $24.6 \pm 1.2\%$ and $14.9 \pm 0.6\%$ compared to the $26.7 \pm 1.4\%$ for Zeo-Y_H (Si/Al = 30:1) under the standard reaction conditions, i.e., 45 bar H_2 , 375 °C, 2 hrs (Table 4, Entries 1-3). As the SAR increases the catalytic activity decreases, which may be associated with their surface acidity, and is in accordance with the literature.²¹ The presence of isomerized and unsaturated products confirm that a monofunctional pathway is in operation (Supplementary Table 3, Entries 2, 3) as would be expected. The differences in activity between the different types of zeolites with a comparable SARs (Si/Al = 20-30) are more considerable, i.e., Zeo-Y_H results in a conversion of $26.7 \pm 1.4\%$ whereas ZSM-5_H results in a conversion of $98.0 \pm 2.0\%$ under the standard reaction conditions (cf. Table 2, Entries 6, 9 with Table 4, Entries 2, 3). Such difference also reveals that the confinements originating from the zeolite topology play a more dominant role than the SAR.⁵⁶ Furthermore, Zeo-Y_H shows selectivity toward higher carbon range products due to the larger pore sizes whereas ZSM-5_H favors the C_{1-4} hydrocarbons which may be attributed to smaller pores, typically referred as shape selectivity and correlated to the zeolite topology.⁵⁷⁻⁵⁹

Ni/Zeo-Y_H [60] and Ni/Zeo-Y_H [80] result in conversions of $5.8 \pm 0.5\%$ and $4.9 \pm 0.7\%$ of nC_{16} under 45 bar H_2 at 375 °C reacting for 2 hrs compared to the conversion of $20.9 \pm 1.3\%$ via Ni/Zeo-Y_H (Table 4, Entries 4-6). Similar trends were observed to that of the main library catalysts, i.e., the activity decreases following the

immobilization of Ni NPs of Ni/Zeo-Y_H [60] and Ni/Zeo-Y_H [80] (Table 4, cf. Entry 2, 3 with 5, 6). The activity of the catalysts follows the order: Ni/Zeo-Y_H > Ni/Zeo-Y_H [60] > Ni/Zeo-Y_H [80], the same order as the unmodified supports, and confirming that C-C bond cleavage principally depending on the acid sites of the silica-alumina support at 375 °C. High degrees of saturated products obtained using Ni/Zeo-Y_H [60] and Ni/Zeo-Y_H [80] confirm a bifunctional pathway (Supplementary Table 3, Entries 5, 6). Both Ru/Zeo-Y_H [60] and Ru/Zeo-Y_H [80] led to quantitative conversion as observed for Ru/Zeo-Y_H at 375 °C (Table 4, Entries 7-9), with methane as the dominant product, confirming the hydrogenolysis pathway is in operation (Supplementary Fig. 18).

Table 1 n-Hexadecane deconstruction with the additional Zeo-Y_H-based catalysts with varying SARs.

n-hexadecane $\xrightarrow[\text{45 bar H}_2, \text{375 } ^\circ\text{C}, \text{2 hrs}]{\text{Cat.}}$ C₁- isoC₁₆

Entry	Catalyst	Conv. (%)	C ₁₋₄ Yield (%)	C ₅₋₁₆ Yield (%)
1	Zeo-Y_H	26.7±1.4	3.3	23.3
2	Zeo-Y_H [60]	24.6±1.2	3.8	20.8
3	Zeo-Y_H [80]	14.9±0.6	2.0	12.8
4	Ni/Zeo-Y_H	20.9±1.3	0.4	20.5
5	Ni/Zeo-Y_H [60]	5.8±0.5	0.9	4.9
6	Ni/Zeo-Y_H [80]	4.9±0.7	0.6	4.3
7	Ru/Zeo-Y_H	99.0±1.0	92.4	6.9
8	Ru/Zeo-Y_H [60]	99.0±1.0	98.0	2.0
9	Ru/Zeo-Y_H [80]	99.0±1.0	97.2	2.8

Reaction conditions: n-hexadecane (1.59 g, 7.0 mmol), catalyst (0.1 g, metal loading = 2.5 wt%), S/C ratio (substrate/catalyst weight ratio) ~16, 45 bar H₂, 375 °C, 2 hrs. * All yields were calculated as the carbon yield and isomerized C₁₆ (isoC₁₆) are considered as products.

Supplementary Figure 1 Product distributions after nC_{16} (1.59 g) deconstructions in the presence of Zeo-Y_H-based catalysts with various Si/Al ratios (0.1 g, metal loading = 2.5 wt%): Zeo-Y_H, Zeo-Y_H [60], Zeo-Y_H [80], Ni/Zeo-Y_H, Ni/Zeo-Y_H [60], Ni/Zeo-Y_H [80], Ru/Zeo-Y_H, Ru/Zeo-Y_H [60], and Ru/Zeo-Y_H [80], 45 bar H_2 , 2 hrs. Note that nC_{12} signal originated from the addition as an internal standard has been suppressed, and the yield of C_{12} products herein is derived from the nC_{16} substrate.

Supplementary Table 1 Degrees of saturation of the liquid products obtained from the deconstruction of *n*-hexadecane.

Entry	Catalyst	Conversion (%)	Saturated (%; $\delta = 0.25-2.0$)	Unsaturated (%; $\delta = 2.0-6.0$)	Aromatic (%; $\delta = 6.0-8.0$)
1	Zeo-Y_H	26.7±1.4	98.8±0.1	1.2±0.1	0.1±0.1
2	Zeo-Y_H [60]	24.6±1.2	98.6±0.1	1.4±0.1	0.1±0.1
3	Zeo-Y_H [80]	14.9±0.6	98.7±0.1	1.3±0.1	0.1±0.1
4	Ni/Zeo-Y_H	20.9±1.3	99.8±0.2	0.2±0.2	0.1±0.1
5	Ni/Zeo-Y_H [60]	5.8±0.5	99.7±0.3	0.3±0.3	0.1±0.1
6	Ni/Zeo-Y_H [80]	4.9±0.7	99.5±0.5	0.5±0.5	0.1±0.1

Reaction conditions: *n*-hexadecane (1.59 g, 7.0 mmol), catalyst (0.1 g, metal loading = 2.5 wt%), 45 bar H₂, 375 °C, 2 hrs. Note that degrees of saturation are defined by the ratio of proton integrations in the ¹H NMR spectra to indicate the adjacent carbon-carbon bonds (saturated: $\delta = 0.25-2.0$, unsaturated: $\delta = 2.0-6.0$ and aromatics: $\delta = 6.0-8.0$) given the C-H and C-C bond exclusivity of hydrocarbons.

Ref. 21

Liu, S., Kots, P. A., Vance, B. C., Danielson, A. & Vlachos, D. G. Plastic waste to fuels by hydrocracking at mild conditions. *Sci. Adv.* **7**, 8283–8304 (2021).

Ref. 56

Alaithan, Z. A., Mallia, G. & Harrison, N. M. Monomolecular Cracking of Propane: Effect of Zeolite Confinement and Acidity. *ACS Omega* **7**, 7531–7540 (2022).

Ref. 57

Smit, B. & Maesen, T. L. M. Towards a molecular understanding of shape selectivity. *Nature* vol. 451 671–678 (2008).

Ref. 58

Smit, B. & Maesen, T. L. M. Molecular simulations of zeolites: Adsorption, diffusion, and shape selectivity. *Chem. Rev.* **108**, 4125–4184 (2008).

Ref. 59

Den Hollander, M. A., Wissink, M., Makkee, M. & Moulijn, J. A. Gasoline conversion: Reactivity towards cracking with equilibrated FCC and ZSM-5 catalysts. *Appl. Catal. A Gen.* **223**, 85–102 (2002).

(2) The ZSM-5 supported Co and Ni catalysts were superior compared with the Y supported Co/Ni catalyst, which need an explanation. In the previous tests such as the results in Ref. 21, the ZSM-5 favors the formation of light molecules (e.g. C1-C4), while the Y zeolite with comparable Si/Al ratios could form liquid products. The Si/Al ratios and topological structure should be discussed separately.

As mentioned above in Reviewer#1-Q1, the Si/Al ratio as a control factor has been studied and discussed further in the revised manuscript. Additional discussions concerning the topological structure of the support have also been included:

The differences in activity among amorphous SiO₂-Al₂O₃, crystalline Zeo-Y_H and crystalline ZSM-5_H at different temperatures (Table 2, Entries 1-9) reveal the significance of local confinements and the corresponding topology of a given silica-alumina material towards the activity and selectivity of the catalyst (see below).^{42,43}

The Co and Ni NPs are unable to cleave C-C bonds efficiently, and therefore the catalytic activity is mainly determined by the acid sites when the reaction is conducted above the onset temperature of the silica-alumina support, evidenced by the proportion of C₃ product for the hydrogenolysis of nC₁₆ using Co/ZSM-5_H or Ni/ZSM-5_H (Fig. 2b-c). Hence, Co/ZSM-5_H and Ni/ZSM-5_H are more active than the Zeo-Y_H-based counterparts, i.e., Co/Zeo-Y_H and Ni/Zeo-Y_H, due to the higher

activity of the ZSM-5_H support, and despite having identical NP coverages. Such interplay between the immobilized NPs and surface is often referred to as the acid-metal balance and can further affect product selectivity, particularly for bifunctional hydrocracking catalysts.^{47,48}

The differences in activity between the different types of zeolites with a comparable SARs (Si/Al = 20-30) are more considerable, i.e., Zeo-Y_H results in a conversion of 26.7±1.4% whereas ZSM-5_H results in a conversion of 98.0±2.0% under the standard reaction conditions (cf. Table 2, Entries 6, 9 with Table 4, Entries 2, 3). Such difference also reveals that the confinements originating from the zeolite topology play a more dominant role than the SAR.⁵⁶ Furthermore, Zeo-Y_H shows selectivity toward higher carbon range products due to the larger pore sizes whereas ZSM-5_H favors the C₁₋₄ hydrocarbons which may be attributed to smaller pores, typically referred as shape selectivity and correlated to the zeolite topology.⁵⁷⁻⁵⁹

Ref. 42

Leydier, F., Chizallet, C., Costa, D. & Raybaud, P. Revisiting carbenium chemistry on amorphous silica-alumina: Unraveling their milder acidity as compared to zeolites. *J. Catal.* **325**, 35–47 (2015).

Ref. 43

Chai, Y., Dai, W., Wu, G., Guan, N. & Li, L. Confinement in a Zeolite and Zeolite Catalysis. *Acc. Chem. Res.* **54**, 2894–2904 (2021).

Ref. 47

Anaya, F., Zhang, L., Tan, Q. & Resasco, D. E. Tuning the acid-metal balance in Pd/ and Pt/zeolite catalysts for the hydroalkylation of *m*-cresol. *J. Catal.* **328**, 173–185 (2015).

Ref. 48

Monteiro, C. A. A., Costa, D., Zotin, J. L. & Cardoso, D. Effect of metal-acid site balance on hydroconversion of decalin over Pt/Beta zeolite bifunctional catalysts. *Fuel* **160**, 71–79 (2015).

(3) The coke formation during the catalysis should be mentioned. As known, the zeolite with strong acidity easily caused the coking in the PE depolymerization at high temperature. This feature would reduce the recyclability of the zeolite-based catalysts for PE depolymerization without further coke-removal treatment. As suggested, we briefly discuss the issue of coke formation for depolymerization in the revised manuscript, citing relevant literature and commenting on why coking was not observed in our studies:

Although significant coking was not observed, presumably due to the short reaction time and the application of hydrogen (Table 2), the use of a solvent is also expected to prevent coke formations and to preserve catalytic performance for more challenging of polymer substrates over longer-term operation,^{65,66} particularly in the cases via highly acidic catalysts.^{67,68} Nonetheless, regeneration processes for coke removal could be applied if required.^{69,70}

Ref. 65

Zachariah, A., Wang, L., Yang, S., Prasad, V. & De Klerk, A. Suppression of coke formation during bitumen pyrolysis. *Energy and Fuels* **27**, 3061–3070 (2013).

Ref. 66

Wan, H., Chaudhari, R. V. & Subramaniam, B. Catalytic hydroprocessing of *p*-cresol: Metal, solvent and mass-transfer effects. in *Topics in Catalysis* vol. 55 129–139 (Springer, 2012).

Ref. 67

Gobin, K. & Manos, G. Polymer degradation to fuels over microporous catalysts as a novel tertiary plastic recycling method. *Polym. Degrad. Stab.* **83**, 267–279 (2004).

Ref. 68

Xian, X. et al. Acidity tuning of HZSM-5 zeolite by neutralization titration for coke inhibition in supercritical catalytic cracking of n-dodecane. *Appl. Catal. A Gen.* **623**, 118278 (2021).

Ref. 69

Nakasaka, Y., Tago, T., Konno, H., Okabe, A. & Masuda, T. Kinetic study for burning regeneration of coked MFI-type zeolite and numerical modeling for regeneration process in a fixed-bed reactor. *Chem. Eng. J.* **207–208**, 368–376 (2012).

Ref. 70

Kassargy, C. et al. Study of the effects of regeneration of USY zeolite on the catalytic cracking of polyethylene. *Appl. Catal. B Environ.* **244**, 704–708 (2019).

(4) How about the reaction channels to form liquid and gaseous products? In the Pt and zeolite tandem catalysis at temperatures lower than 300 C, the hydrogenolysis is regarded for the C-C cleavage, while the zeolite catalyzes the secondary cracking reactions to optimize the product selectivity. The Co and Ni would have lower activity than Pt for the hydrogenolysis, and zeolite catalyzed cracking might be dominant for PE cracking at high temperature (e.g. 375 C). The functions of Co/Ni metals and zeolite should be described in the revised manuscript.

The discussion about reaction channels and the function of the Co and Ni NPs together with relevant citations has been expanded. Note that the new text given below is better appreciated where it is embedded within the manuscript:

The Co and Ni NPs are unable to cleave C-C bonds efficiently, and therefore the catalytic activity is mainly determined by the acid sites when the reaction is conducted above the onset temperature of the silica-alumina support, evidenced by the proportion of C₃ product for the hydrogenolysis of nC₁₆ using Co/ZSM-5_H or Ni/ZSM-5_H (Fig. 2b-c).

This difference is because the Co and Ni NPs efficiently catalyze the hydrogenation of unsaturated bonds formed during the cracking process.

Moreover, the Co and Ni metal sites appear to provide an alternative route facilitating the dissociation of intermediates away from the acid sites that would otherwise be cleaved further by them to afford higher degrees of saturated products in higher carbon ranges.

Note that evaluating metal NPs dispersed on an inactive support material may provide insights on the onset temperature for C-C bond hydrogenolysis of a given metal. Ru NPs immobilized on carbon (Ru/C) were shown to depolymerize PE via a hydrogenolysis mechanism with an onset temperature around 220 °C, whereas Pt/C, Pd/C, and Rh/C require temperature ≥280 °C.^{23,24} As Ru NPs efficiently catalyze C-C bond hydrogenolysis, the activity and selectivity are dominated by the Ru NPs, especially where the silica-alumina support is less active, e.g., SiO₂-Al₂O₃, and below the onset temperature of the support.

With the exception of Ru, the influence of a metal on the selectivity of the reaction is less predictable, especially when the hydrogenolysis onset temperature is close to the temperature required by the support to initiate hydrocracking. Thus, the overhead of reaction temperature with respect to hydrogenolysis/hydrocracking on set temperature critically influences which mechanism prevails. For instance, ultra-stable Y zeolite and beta zeolite modified with Pt NPs (i.e., ≥280 °C for hydrogenolysis) behave as bifunctional hydrocracking catalysts at around 300 °C.⁵⁵

Ref. 23

Rorrer, J. E., Beckham, G. T. & Román-Leshkov, Y. Conversion of Polyolefin Waste to Liquid Alkanes with Ru-Based Catalysts under Mild Conditions. *JACS Au* **1**, 8–12 (2021).

Ref. 24

Jia, C. et al. Deconstruction of high-density polyethylene into liquid hydrocarbon fuels and lubricants by hydrogenolysis over Ru catalyst. *Chem Catal.* **1**, 437–455 (2021).

Ref. 55

Bin Jumah, A., Anbumuthu, V., Tedstone, A. A. & Garforth, A. A. Catalyzing the Hydrocracking of Low Density Polyethylene. *Ind. Eng. Chem. Res.* **58**, 20601–20609 (2019).

(5) Carbon balance values are suggested to be included in Figure 2 and Table 2.

As we do not observe significant coke formation in the reactions involving nC₁₆, see Figure 2 and Table 2, due to the use of a liquid substrate, short reaction time and hydrogen atmosphere, we assume a quantitative carbon balance.

Reviewer #2 (Remarks to the Author):

This paper studies hydrogenolysis of n-hexadecane in a batch reactor as a model polyolefin feed. The key learning from this paper is shown in Figure 4 where the author plot the activity of several different catalysts. The authors have tested several catalysts and combine it in this figure. Several recent studies have been done on hydrogenolysis of plastics into an oil mixture including the work of Y Roman and D Vlachos. I do not see any new scientific learnings about the catalysis from this study compared to those other studies and am concerned about the novelty of this work. Hydrogenolysis is going to be an expensive reaction to do because of the high hydrogen requirements and the low value of the products that are produced (mainly fuels or lubricants). I recommend the authors report their hydrogen consumption. I am not sure why they did not start with an alkene rather than an alkane as pyrolysis of polyolefins produces alkenes as the primary product. There has been lots of work on conversion of polyolefins over zeolite catalysts. I am however not sure what is learned from this study that is not already known in the literature.

We have modified the Abstract, parts of the Introduction and Conclusions to further articulate the novel aspects and core values of the manuscript. Please refer directly to the manuscript where the revised text is highlighted within each section mentioned above to better appreciate those modifications. Relevant studies by Y. Roman and D. Vlachos have also been added as references and integrated into the revised manuscript. We believe that the additional experiments and discussions suggested by all three reviewers enhance what is learned from this study. We appreciate the perspective of Reviewer#2 concerning the cost of hydrogen, and consequently highlight the benefits of hydrogenolysis/hydrocracking of waste polymers in comparison to alternative approaches in the Conclusions. We can estimate the hydrogen consumption, however, it is not a common practice in the literature which has been referred to (Ref. 22-25) in the manuscript. Besides, due to the larger expected errors originating from the nature of a different detector for hydrogen detection than light hydrocarbons, we decided against including this parameter in the original manuscript. See also the highlighted revisions for the decision of using an alkane as the model in the revised manuscript – only the additional references are listed below:

Ref. 23

Rorrer, J. E., Beckham, G. T. & Román-Leshkov, Y. Conversion of Polyolefin Waste to Liquid Alkanes with Ru-Based Catalysts under Mild Conditions. *JACS Au* **1**, 8–12 (2021).

Ref. 25

Kots, P. A. et al. Polypropylene Plastic Waste Conversion to Lubricants over Ru/TiO₂ Catalysts. *ACS Catal.* **11**, 8104–8115 (2021).

Ref. 76

Thiounn, T. & Smith, R. C. Advances and approaches for chemical recycling of plastic waste. *Journal of Polymer Science* vol. 58 1347–1364 (2020).

Ref. 77

Rahimi, A. R. & Garcíá, J. M. Chemical recycling of waste plastics for new materials production. *Nature Reviews Chemistry* vol. 1 1–11 (2017).

Ref. 78

Ragaert, K., Delva, L. & Van Geem, K. Mechanical and chemical recycling of solid plastic waste. *Waste Management* vol. 69 24–58 (2017).

Ref. 79

Das, S. et al. Solid waste management: Scope and the challenge of sustainability. *J. Clean. Prod.* **228**, 658–678 (2019).

Ref. 80

Cucchiella, F., D'Adamo, I. & Gastaldi, M. Sustainable waste management: Waste to energy plant as an alternative to landfill. *Energy Convers. Manag.* **131**, 18–31 (2017).

Ref. 81

Statista. Forecasted breakdown of renewable hydrogen production costs worldwide from 2020 to 2030. <https://www.statista.com/statistics/1220812/global-hydrogen-production-cost-forecast-by-scenario/> (2021).

Ref. 82

IEA. *The Future of Hydrogen*.

<https://www.iea.org/reports/the-future-of-hydrogen> (2019).

Ref. 83

Yan, Q. et al. A comprehensive review on selective catalytic reduction catalysts for NO_x emission abatement from municipal solid waste incinerators. *Environ. Prog. Sustain. Energy* **35**, 1061–1069 (2016).

Ref. 84

Tseng, H. H., Wey, M. Y., Liang, Y. S. & Chen, K. H. Catalytic removal of SO₂, NO and HCl from incineration flue gas over activated carbon-supported metal oxides. *Carbon N. Y.* **41**, 1079–1085 (2003).

Ref. 85

Hart, A., Leeke, G., Greaves, M. & Wood, J. Down-hole heavy crude oil upgrading by CAPRI: Effect of hydrogen and methane gases upon upgrading and coke formation. *Fuel* **119**, 226–235 (2014).

Here are a few other comments:

(1) What is the formula for yield?

The formula for calculating yield has now been included in the revised manuscript:

$$\begin{aligned} C_{k(k=5-16)} \text{ Yield (\%)} &= \frac{n_{(C_k, \text{ act.})}}{n_{(C_k, \text{ theo. max})}} * 100 \\ &= \frac{\text{Coef} \cdot (\text{ECN}) * \frac{\text{Area}_{(C_k, \text{ act.})}}{\text{Area}_{(nC_{12}, \text{ int. std.})}} * n_{(nC_{12}, \text{ int. std.})}}{n_{(C_k, \text{ theo. max})}} * 100 \dots (1) \end{aligned}$$

$$\begin{aligned} C_{l(l=1-4)} \text{ Yield (\%)} &= \frac{n_{(C_l, \text{ act.})}}{n_{(C_l, \text{ theo. max})}} * 100 \\ &= \frac{\frac{\text{Area}_{(C_l, \text{ act.})}}{\text{Area}_{(C_l, \text{ ref. gas})}} * \frac{RT}{P_f V_{(\text{autoclave})}}}{n_{(C_l, \text{ theo. max})}} * 100 \dots (2) \end{aligned}$$

(2) Why are the error bars on Ni/Co yields are much larger compared to the yield using ZSM-5 in Figure 2? Some of the error bars are almost as big as the yield of some products.

Some of the errors at short reaction times (2 hrs) are large given the reactions were under the relative sensitive region, i.e., in the middle of a sigmoidal curve, but they become much smaller at longer reaction times (6 hrs) as the reactions moved toward the end of the sigmoidal curve. Nonetheless, we would hope that they are acceptable, but if essential we can repeat the experiments.

(3) More evidence/discussion is required to classify the reaction mechanism. I do not think classifying by product distribution is sufficient.

We have added considerably more details and the suggested discussions from all three reviews in the revised manuscript that should further supplement the approach used to classify mechanisms according to product distributions. Moreover, we have slightly softened the language so that we do not overstate the utility of our approach upon the complexity of all the parameters that define the overall reaction mechanism.

(4) Discussion about the confinement effects should be added.

The discussions about confinement effects have now been included with additional literature cited in the revised manuscript:

The differences in activity among amorphous SiO₂-Al₂O₃, crystalline Zeo-Y_H and crystalline ZSM-5_H at different temperatures (Table 2, Entries 1-9) reveal the significance of local confinements and the corresponding topology of a given silica-alumina material towards the activity and selectivity of the catalyst (see below).^{42,43}

The differences in activity between the different types of zeolites with a comparable SARs (Si/Al = 20-30) are more considerable, i.e., Zeo-Y_H results in a conversion of 26.7±1.4% whereas ZSM-5_H results in a conversion of 98.0±2.0% under the standard reaction conditions (cf. Table 2, Entries 6, 9 with Table 4, Entries 2, 3). Such difference also reveals that the confinements originating from the zeolite topology play a more dominant role than the SAR.⁵⁶

Ref. 42

Leydier, F., Chizallet, C., Costa, D. & Raybaud, P. Revisiting carbenium chemistry on amorphous silica-alumina: Unraveling their milder acidity as compared to zeolites. *J. Catal.* **325**, 35–47 (2015).

Ref. 43

Chai, Y., Dai, W., Wu, G., Guan, N. & Li, L. Confinement in a Zeolite and Zeolite Catalysis. *Acc. Chem. Res.* **54**, 2894–2904 (2021).

Ref. 56

Alaithan, Z. A., Mallia, G. & Harrison, N. M. Monomolecular Cracking of Propane: Effect of Zeolite Confinement and Acidity. *ACS Omega* **7**, 7531–7540 (2022).

(5) Proximity of acid sites and metal sites should also be included. I do not expect an extensive analysis but at least some demonstration is necessary.

We performed further experiments to demonstrate the impact of the proximity of acid sites and metal sites that are discussed together with additional literature in the revised manuscript:

The proximity of the acid sites and metal sites was investigated by comparing the products obtained using ZSM-5_H + Ni/γAl₂O₃ (Ni-modified gamma-alumina prepared using identical procedure as Ni/ZSM-5_H) and Ni/ZSM-5_H + γAl₂O₃ as catalysts.⁵² ZSM-5_H + Ni/γAl₂O₃ may be considered as a low intimacy combination and Ni/ZSM-5_H + γAl₂O₃ as a high intimacy combination. The low intimacy combination resulted in near-quantitative conversion with a unimodal distribution indicative of a monofunctional pathway typical of ZSM-5_H (cf. Fig. 2a-left with 2e-left), whereas the high intimacy combination led to a conversion of 93.9±0.3% with a bimodal distribution similar to the bifunctional pathway observed for Ni/ZSM-5_H (cf. Fig. 2c-left with 2e-right). These results confirm the importance of acid and metal site proximity in influencing the product distribution as reported elsewhere.^{39,52,53} Note that γAl₂O₃ shows some activity (conv. = 6.8%±0.9) in the transformation of nC₁₆ at 375 °C, explaining the slightly higher activity of the Ni/ZSM-5_H + γAl₂O₃ combination

compared to Ni/ZSM-5_H alone. Moreover, the Co and Ni metal sites appear to provide an alternative route facilitating the dissociation of intermediates away from the acid sites that would otherwise be cleaved further by them to afford higher degrees of saturated products in higher carbon ranges.

Fig. 2e *n*C₁₆ (1.59 g, 7.0 mmol) deconstruction using Ni/ZSM-5_H (0.1 g, metal loading = 2.5 wt%) + γ Al₂O₃ (0.1 g) and ZSM-5_H (0.1 g) + Ni/ γ Al₂O₃ (0.1 g, metal loading = 2.5 wt%), 45 bar H₂, 2 hrs.

Ref. 39

Oenema, J. et al. Influence of Nanoscale Intimacy and Zeolite Micropore Size on the Performance of Bifunctional Catalysts for *n*-Heptane Hydroisomerization. *ACS Catal.* **10**, 14245–14257 (2020).

Ref. 52

Mendes, P. S. F., Silva, J. M., Ribeiro, M. F., Daudin, A. & Bouchy, C. Bifunctional Intimacy and its Interplay with Metal-Acid Balance in Shaped Hydroisomerization Catalysts. *ChemCatChem* **12**, 4582–4592 (2020).

Ref. 53

Zhang, Y. et al. Hydroisomerization of *n*-dodecane over bi-porous Pt-containing bifunctional catalysts: Effects of alkene intermediates' journey distances within the zeolite micropores. *Fuel* **236**, 428–436 (2019).

Reviewer #3 (Remarks to the Author):

Dyson et al report the conversion of a low molecular weight polyethylene into liquid and gaseous hydrocarbon products with a series of supported metal catalysts on acid supports (amorphous silica-alumina and zeolites). In general, the manuscript is well written and organized, the concept is very interesting and firmly examined. In summary, the work presented here merits publication in Nat. Commun. and will certainly be of interest to the readership in this journal. However, there are several specific questions for this manuscript and they are listed below.

(1) Where is the hydrogen coming from in the metal-free reactions with N₂?

The source of hydrogen has been described in the revised manuscript with an appropriate reference given:

Despite the absence of an external hydrogen source when the reaction is carried out under a nitrogen atmosphere, the hydrogen atoms in the hydrocarbon substrate are able to be transferred, to afford saturated products together with the corresponding unsaturated products resulting from the hydrogen transfer.⁴⁴

Ref. 44

*Shimada, I., Takizawa, K., Fukunaga, H., Takahashi, N. & Takatsuka, T. Catalytic cracking of polycyclic aromatic hydrocarbons with hydrogen transfer reaction. Fuel **161**, 207–214 (2015).*

(2) Does the degree of unsaturation from the final product account for how much hydrogen would need to be consumed to convert the reactant to C1 – C8s?

Sufficient hydrogen (45 bar) was used to produce methane quantitatively and this is now highlighted in the revised text:

Each catalyst was evaluated under sufficient H₂ (~1.15 eq.) to quantitatively produce methane...

(3) Would be interesting if the authors varied H₂ pressure and what effect that would have on unsaturated product formation, since they claim a benefit of the bifunctional pathway is suppressing coke precursors.

Different applied hydrogen pressures (30 bar and 60 bar) were studied in addition to the original pressure of 45 bar used. The results and discussion have been included in the revised manuscript and Supplementary Information:

Further control experiments using nC₁₆ as a substrate under different hydrogen pressures revealed that the degrees of saturated liquid hydrocarbons increase along the increased pressure (i.e., the amount of H₂ increases). When ZSM-5-H is applied as the catalyst, the percentage of saturated products increases from 86.5% under 30 bar of H₂ to 88.1% under 60 bar of H₂ (corresponding to ~0.80 and ~1.55 eq. required to quantitatively produce methane, respectively, Table 3, Entries 6,7, and Supplementary Fig. 17). With Ni/ZSM-5_H, 98.6% and 99.9% of saturated products are obtained at 30 and 60 bar of H₂, respectively (Table 3, Entries 8,9, and Supplementary Fig. 18). The higher the hydrogen pressure shows the higher the percentage of saturated products, though only to a modest extent. In contrast, the metal modification has a greater influence, significantly increasing the amount of saturated products in the final product distribution.

Table 2 Degrees of saturation of the liquid products obtained from the deconstruction of n-hexadecane *under hydrogen*.

Entry	Catalyst	H ₂ (bar)	Time (hr)	Conv. (%)	Saturated (%, $\delta = 0.25-2.0$)	Unsaturated (%, $\delta = 2.0-6.0$)	Aromatic (%, $\delta = 6.0-8.0$)
1	ZSM-5_H	45	2	98.0±2.0	87.4±4.3	7.9±2.6	4.8±1.9
2	Co/ZSM-5_H	45	2	49.0±3.7	99.4±0.5	0.5±0.4	0.1±0.1
3	Ni/ZSM-5_H	45	2	85.6±4.1	97.6±0.8	2.3±0.7	0.1±0.1
4	Co/ZSM-5_H	45	6	93.1±4.3	96.2±1.8	3.1±1.6	0.6±0.2
5	Ni/ZSM-5_H	45	6	99.8±0.2	99.9±0.1	0.1±0.1	0.1±0.1
6	ZSM-5_H	30	2	99.8±0.1	86.5±0.5	8.8±0.5	4.7±0.1
7	ZSM-5_H	60	2	99.9±0.1	88.1±0.2	7.4±0.2	4.5±0.1
8	Ni/ZSM-5_H	30	2	99.5±0.5	98.6±0.8	1.0±0.7	0.4±0.1
9	Ni/ZSM-5_H	60	2	94.9±2.2	99.9±0.1	0.1±0.1	0.1±0.1

Reaction conditions: n-hexadecane (1.59 g, 7.0 mmol), catalyst (0.1 g, metal loading = 2.5 wt%), 375 °C. Note that degrees of saturation are defined by the ratio of proton integrations in ¹H NMR spectra to indicate the adjacent carbon-carbon bonds (saturated: $\delta = 0.25-2.0$, unsaturated: $\delta = 2.0-6.0$ and aromatics: $\delta = 6.0-8.0$) given the C-H and C-C bond exclusivity of hydrocarbons.

a nC₁₆ deconstruction:
ZSM-5_H, 30 bar H₂, 375 °C, 2 hrs

b nC₁₆ deconstruction:
ZSM-5_H, 60 bar H₂, 375 °C, 2 hrs

Supplementary Figure 2 ¹H NMR spectra of liquid products after nC₁₆ (1.59 g) deconstruction in the presence of ZSM-5_H (0.1 g) 375 °C, 2 hrs under **a** 30 bar H₂, proton integration: 1.00, 9.16, 16.03, 148.03 = CDCl₃ (gray marked area, δ = 7.20-7.35), aromatic (δ = 6.0-8.0), unsaturated (δ = 2.0-6.0), saturated (δ = 0.25-2.0) and **b** 60 bar H₂, proton integration: 1.00, 8.15, 11.58, 141.13 = CDCl₃ (gray marked area, δ = 7.20-7.35), aromatic (δ = 6.0-8.0), unsaturated (δ = 2.0-6.0), saturated (δ = 0.25-2.0).

Supplementary Figure 3 ¹H NMR spectra of liquid products after nC₁₆ (1.59 g) deconstruction in the presence of Ni/ZSM-5_H (0.1 g) 375 °C, 2 hrs under **a** 30 bar H₂, proton integration: 1.00, 3.74, 2.40, 913.45 = CDCl₃ (gray marked area, δ = 7.20-7.35), aromatic (δ = 6.0-8.0), unsaturated (δ = 2.0-6.0), saturated (δ = 0.25-2.0) and **b** 60 bar H₂, proton integration: 1.00, 1.01, 0.83, 1077.34 = CDCl₃ (gray marked area, δ = 7.20-7.35), aromatic (δ = 6.0-8.0), unsaturated (δ = 2.0-6.0), saturated (δ = 0.25-2.0).

(4) What about the bifunctional catalysts make C3 and C5/C6 the major peaks in a bimodal distribution? Why is C4 an unlikely product here?

We have revised the manuscript to address this issue:

The Co and Ni NPs are unable to cleave C-C bonds efficiently, and therefore the catalytic activity is mainly determined by the acid sites when the reaction is conducted above the onset temperature of the silica-alumina support, evidenced by the proportion of C₃ product for the hydrogenolysis of nC₁₆ using Co/ZSM-5_H or Ni/ZSM-5_H (Fig. 2b-c). Hence, Co/ZSM-5_H and Ni/ZSM-5_H are more active than the Zeo-Y_H-based counterparts, i.e., Co/Zeo-Y_H and Ni/Zeo-Y_H, due to the higher activity of the ZSM-5_H support, and despite having identical NP coverages. Such interplay between the immobilized NPs and surface is often referred to as the acid-metal balance and can further affect product selectivity, in these cases not specifically favoring C₄ products.^{47,48}

Ref. 47

Anaya, F., Zhang, L., Tan, Q. & Resasco, D. E. Tuning the acid-metal balance in Pd/ and Pt/zeolite catalysts for the hydroalkylation of m-cresol. *J. Catal.* 328, 173–185 (2015).

Ref. 48

Monteiro, C. A. A., Costa, D., Zotin, J. L. & Cardoso, D. Effect of metal-acid site balance on hydroconversion of decalin over Pt/Beta zeolite bifunctional catalysts. *Fuel* 160, 71–79 (2015).

(5) Having a scheme showcasing these reactions and discussing why certain products are/are not favorable would make this argument easier to follow. The authors map the catalysts according to their hydrogenolysis vs. hydrocracking activity, and so being able to visualize specific mechanisms responsible for each transformation and how each catalyst falls in line with a scaling combination of both would make more sense with the product distribution given. The discussion about hydrogenolysis and hydrocracking mechanisms has been expanded to include typical key intermediate species and these species have been incorporated in the scheme in Table 1. We hope that this satisfies the reviewer's suggestion to better visualize why a certain mechanism favors a specific product/product distribution.

The formation of methane is considered an important indicator of a hydrogenolysis mechanism with alkylidyne intermediates, as C₁ intermediates such as the methenium cation are disfavored in (hydro)cracking mechanisms.²²

Ref. 22

Weitkamp, J. *Catalytic Hydrocracking-Mechanisms and Versatility of the Process*. *ChemCatChem* **4**, 292–306 (2012).

REVIEWER COMMENTS

Reviewer #1 (Remarks to the Author):

Authors should calculate the carbon balance before and after the reaction. After this modification, this manuscript might be accepted for the publication.

Reviewer #2 (Remarks to the Author):

The authors have adequately responded to all of my previous concerns except my comment about reporting the hydrogen consumption. While the hydrogen consumption has not been reported in previous hydrotreating plastic papers it is a common number reported in the hydrotreating of vacuum gas oil and in biomass. My opinion is that the hydrogen consumption should be reported in all studies when hydrogen is used as a feed. I do understand that it can be hard to estimate but it would be helpful when technologies are evaluated.

Reviewer #3 (Remarks to the Author):

The authors have sufficiently addressed the previous issues and the current manuscript is appropriate for publication. However, i find very strange that the Authors did not properly cite pioneering works in plastic conversion from the A. Sadow, S. Scott, M. Delferro, K. Poeppelmeier, and W. Huang groups, but mention only papers from Roman and Vlachos groups that appeared in the literature after the ones cited above.

Reviewer #1 (Remarks to the Author):

Authors should calculate the carbon balance before and after the reaction. After this modification, this manuscript might be accepted for the publication.

To determine the carbon balance TGA of selected catalysts was conducted confirming a well-closed carbon balance of the gaseous and liquid products. The carbon balance has now been provided in the form of weight and the manuscript and supporting information modified as outlined below

Significant coking was not observed presumably due to the short reaction time and the application of hydrogen (Tables 2 and 4), evidenced by the similar curves obtained from thermogravimetric analysis (TGA) of fresh and used catalysts (Supplementary Fig. 21). No apparent differences in TGA curves above 500 °C were observed, indicating a well-closed carbon balance by the gaseous and liquid products due to limited solid residual formation during reaction.

The results from the main library catalysts are summarized in Table 2 and Supplementary Figs. 12-15 provide detailed product distributions, and Supplementary Table 3 provides the carbon balance in weight and hydrogen consumption.

Zeo-Y_H-based catalysts with different Si/Al ratios (SARs) were also evaluated in the deconstructions of nC_{16} at 375 °C, see Table 4 and Supplementary Fig. 20 for the full product distributions, and Supplementary Table 4 for the carbon balance in weight and hydrogen consumption.

Supplementary Figure 21 Thermogravimetric analysis of fresh and used catalyst of **a** Zeo-Y_H **b** Zeo-Y_H [80] **c** ZSM-5_H **d** Ni/ZSM-5_H **e** Ru/ZSM-5_H with a ramping rate of 5 °C/min from 35 to 900 °C and a flow rate of 20 mL/min under air.

Supplementary Table 1 *n*-Hexadecane deconstructions with the 12 catalysts from the main library at 275, 325, and 375 °C including the carbon balance and hydrogen consumption.

Entry	Catalyst	Temp. (°C)	Conv. (%) [g]	C ₁₋₄ Yield (%) [g]	C ₅₋₁₆ Yield (%) [g]	H ₂ Cons. (%) [g]
1	SiO ₂ -Al ₂ O ₃	275 °C	2.1±0.2 [0.03±0.00]	0.0 [0.00]	2.1 [0.03]	1 [0.00]
2	SiO ₂ -Al ₂ O ₃	325 °C	3.8±1.0 [0.06±0.02]	0.1 [0.00]	3.7 [0.06]	6 [0.01]
3	SiO ₂ -Al ₂ O ₃	375 °C	2.1±0.2 [0.03±0.00]	0.1 [0.00]	2.0 [0.03]	5 [0.01]
4	Zeo-Y_H	275 °C	4.5±0.3 [0.07±0.00]	0.3 [0.01]	4.2 [0.07]	2 [0.00]
5	Zeo-Y_H	325 °C	8.5±0.5 [0.14±0.01]	0.7 [0.01]	7.7 [0.12]	10 [0.02]
6	Zeo-Y_H	375 °C	26.7±1.4 [0.42±0.02]	3.3 [0.05]	23.3 [0.37]	11 [0.03]
7	ZSM-5_H	275 °C	13.7±2.4 [0.22±0.04]	3.8 [0.06]	10.0 [0.16]	5 [0.01]
8	ZSM-5_H	325 °C	91.6±4.4 [1.46±0.07]	35.5 [0.58]	56.2 [0.90]	17 [0.04]
9	ZSM-5_H	375 °C	98.0±2.0 [1.56±0.03]	77.3 [1.27]	20.8 [0.33]	23 [0.06]
10	Co/SiO ₂ -Al ₂ O ₃	275 °C	2.3±0.2 [0.04±0.00]	0.1 [0.00]	2.2 [0.03]	4 [0.01]
11	Co/SiO ₂ -Al ₂ O ₃	325 °C	2.3±0.3 [0.04±0.00]	0.0 [0.00]	2.2 [0.03]	2 [0.00]
12	Co/SiO ₂ -Al ₂ O ₃	375 °C	2.4±0.1 [0.04±0.00]	0.1 [0.00]	2.2 [0.03]	7 [0.02]
13	Co/Zeo-Y_H	275 °C	1.9±0.4 [0.03±0.01]	0.0 [0.00]	1.9 [0.03]	4 [0.01]
14	Co/Zeo-Y_H	325 °C	5.5±1.2 [0.09±0.02]	0.2 [0.00]	5.2 [0.08]	6 [0.01]
15	Co/Zeo-Y_H	375 °C	6.1±1.2 [0.10±0.02]	1.2 [0.02]	4.9 [0.08]	8 [0.02]
16	Co/ZSM-5_H	275 °C	1.9±0.1 [0.03±0.00]	0.1 [0.00]	1.8 [0.03]	6 [0.01]
17	Co/ZSM-5_H	325 °C	5.9±0.1 [0.09±0.00]	1.2 [0.02]	4.7 [0.07]	6 [0.01]
18	Co/ZSM-5_H	375 °C	49.0±3.7 [0.78±0.06]	12.7 [0.21]	36.3 [0.58]	13 [0.03]
19	Ni/SiO ₂ -Al ₂ O ₃	275 °C	2.6±0.6 [0.04±0.01]	0.1 [0.00]	2.6 [0.04]	3 [0.01]
20	Ni/SiO ₂ -Al ₂ O ₃	325 °C	3.3±1.4 [0.05±0.02]	0.1 [0.00]	3.2 [0.05]	4 [0.01]
21	Ni/SiO ₂ -Al ₂ O ₃	375 °C	3.8±1.2 [0.06±0.02]	0.2 [0.00]	3.7 [0.06]	7 [0.02]
22	Ni/Zeo-Y_H	275 °C	2.1±0.6 [0.03±0.01]	0.6 [0.01]	1.5 [0.02]	4 [0.01]
23	Ni/Zeo-Y_H	325 °C	4.4±0.4 [0.07±0.01]	0.4 [0.01]	4.0 [0.06]	5 [0.01]
24	Ni/Zeo-Y_H	375 °C	20.9±1.3 [0.33±0.02]	0.4 [0.01]	20.5 [0.31]	8 [0.02]
25	Ni/ZSM-5_H	275 °C	2.2±0.2 [0.03±0.00]	0.3 [0.00]	1.9 [0.03]	4 [0.01]
26	Ni/ZSM-5_H	325 °C	8.2±1.8 [0.13±0.03]	0.5 [0.01]	7.7 [0.12]	7 [0.02]
27	Ni/ZSM-5_H	375 °C	85.6±4.1 [1.36±0.07]	28.2 [0.46]	57.3 [0.92]	13 [0.03]
28	Ru/SiO ₂ -Al ₂ O ₃	275 °C	95.6±4.0 [1.52±0.06]	41.5 [0.71]	54.2 [0.86]	27 [0.07]
29	Ru/SiO ₂ -Al ₂ O ₃	325 °C	98.5±0.3 [1.57±0.00]	75.8 [1.33]	22.6 [0.36]	60 [0.14]
30	Ru/SiO ₂ -Al ₂ O ₃	375 °C	99.8±0.2 [1.59±0.00]	96.6 [1.72]	3.3 [0.05]	83 [0.20]
31	Ru/Zeo-Y_H	275 °C	96.0±0.9 [1.53±0.01]	56.6 [0.99]	39.3 [0.63]	44 [0.11]
32	Ru/Zeo-Y_H	325 °C	99.6±0.3 [1.58±0.00]	91.5 [1.62]	8.3 [0.13]	73 [0.18]
33	Ru/Zeo-Y_H	375 °C	99.0±1.0 [1.57±0.02]	92.4 [1.65]	6.9 [0.11]	85 [0.21]
34	Ru/ZSM-5_H	275 °C	99.4±0.6 [1.58±0.01]	92.8 [1.65]	6.6 [0.10]	54 [0.13]
35	Ru/ZSM-5_H	325 °C	98.0±2.0 [1.56±0.03]	99.7 [1.78]	0.3 [0.01]	88 [0.21]
36	Ru/ZSM-5_H	375 °C	99.8±0.2 [1.59±0.00]	98.8 [1.77]	1.1 [0.02]	88 [0.21]

Reaction conditions: *n*-hexadecane (1.59 g, 7.0 mmol), catalyst (0.1 g, metal loading = 2.5 wt%), S/C ratio (substrate/catalyst weight ratio) ~16, 45 bar H₂, 2 hrs. * All yields were calculated as the carbon yield and isomerized C₁₆ (isoC₁₆) are considered as products. Note that ~87% H₂ consumption (~105.0 mmol) is able to produce methane quantitatively due to the ~1.15 eq. H₂ stoichiometry.

Supplementary Table 2 *n*-Hexadecane deconstruction using the Zeo-Y_H-based catalysts with varying SARs including the carbon balance and hydrogen consumption.

Entry	Catalyst	Conv. (%) [g]	C ₁₋₄ Yield (%) [g]	C ₅₋₁₆ Yield (%) [g]	H ₂ Cons. (%) [g]
1	Zeo-Y _H	26.7±1.4 [0.42±0.02]	3.3 [0.05]	23.3 [0.37]	11 [0.03]
2	Zeo-Y _H [60]	24.6±1.2 [0.39±0.02]	3.8 [0.06]	20.8 [0.33]	8 [0.02]
3	Zeo-Y _H [80]	14.9±0.6 [0.24±0.01]	2.0 [0.03]	12.8 [0.20]	6 [0.01]
4	Ni/Zeo-Y _H	20.9±1.3 [0.33±0.02]	0.4 [0.01]	20.5 [0.31]	8 [0.02]
5	Ni/Zeo-Y _H [60]	5.8±0.5 [0.09±0.01]	0.9 [0.01]	4.9 [0.08]	7 [0.02]
6	Ni/Zeo-Y _H [80]	4.9±0.7 [0.08±0.01]	0.6 [0.01]	4.3 [0.07]	4 [0.01]
7	Ru/Zeo-Y _H	99.0±1.0 [1.57±0.02]	92.4 [1.65]	6.9 [0.11]	85 [0.21]
8	Ru/Zeo-Y _H [60]	99.0±1.0 [1.57±0.02]	98.0 [1.76]	2.0 [0.02]	86 [0.21]
9	Ru/Zeo-Y _H [80]	99.0±1.0 [1.57±0.02]	97.2 [1.75]	2.8 [0.03]	87 [0.21]

Reaction conditions: *n*-hexadecane (1.59 g, 7.0 mmol), catalyst (0.1 g, metal loading = 2.5 wt%), S/C ratio (substrate/catalyst weight ratio) ~16, 45 bar H₂, 375 °C, 2 hrs. * All yields were calculated as the carbon yield and isomerized C₁₆ (isoC₁₆) are considered as products. Note that ~87% H₂ consumption (~105.0 mmol) is able to produce methane quantitatively due to the ~1.15 eq. H₂ stoichiometry.

Reviewer #2 (Remarks to the Author):

The authors have adequately responded to all of my previous concerns except my comment about reporting the hydrogen consumption. While the hydrogen consumption has not been reported in previous hydrotreating plastic papers it is a common number reported in the hydrotreating of vacuum gas oil and in biomass. My opinion is that the hydrogen consumption should be reported in all studies when hydrogen is used as a feed. I do understand that it can be hard to estimate but it would be helpful when technologies are evaluated.

Estimates of hydrogen consumption have been included, see below:

The results from the main library catalysts are summarized in Table 2 and Supplementary Figs. 12-15 provide detailed product distributions, and Supplementary Table 3 provides the carbon balance in weight and hydrogen consumption.

Zeo-Y_H-based catalysts with different Si/Al ratios (SARs) were also evaluated in the deconstructions of nC_{16} at 375 °C, see Table 4 and Supplementary Fig. 20 for the full product distributions, and Supplementary Table 4 for the carbon balance in weight and hydrogen consumption.

Please also see Supplementary Tables 3 and 4 given above.

Reviewer #3 (Remarks to the Author):

The authors have sufficiently addressed the previous issues and the current manuscript is appropriate for publication. However, i find very strange that the Authors did not properly cite pioneering works in plastic conversion from the A. Sadow, S. Scott, M. Delferro, K. Poeppelmeier, and W. Huang groups, but mention only papers from Roman and Vlachos groups that appeared in the literature after the ones cited above.

Key studies from the research groups mentioned by the reviewer have been included in the revised manuscript with the following text:

Pt NPs immobilized on metal oxide (SrTiO₃) or fabricated into a mesoporous shell/active site/core structure (mSiO₂/Pt/SiO₂) have also been used in the hydrogenolysis of PE to afford wax-lubricant range products (C₁₈₋₄₀₊) under 9-14 bar H₂ at temperatures ranging from 250 to 300 °C reacting for 24-96 hrs.²⁶⁻²⁸

Ref. 26

*Wu, X. et al. Size-Controlled Nanoparticles Embedded in a Mesoporous Architecture Leading to Efficient and Selective Hydrogenolysis of Polyolefins. J. Am. Chem. Soc. **144**, 5323–5334 (2022)*

Ref. 27

*Tennakoon, A. et al. Catalytic upcycling of high-density polyethylene via a processive mechanism. Nat. Catal. **3**, 893–901 (2020).*

Ref. 28

*Celik, G. et al. Upcycling Single-Use Polyethylene into High-Quality Liquid Products. ACS Cent. Sci. **5**, 1795–1803 (2019).*

REVIEWERS' COMMENTS

Reviewer #1 (Remarks to the Author):

After the modifications, this work might be accepted for the publication.

Reviewer #2 (Remarks to the Author):

The authors have adequately responded to all of my concerns and I recommend the paper for publication.